# Capillary-associated microglia regulate vascular structure and function through PANX1-P2RY12 coupling in mice

Kanchan Bisht[1,2,10], Kenneth A. Okojie[1,2,10], Kaushik Sharma [1,2,10], Dennis H. Lentferink [1,2], Yu-Yo Sun[1,2], Hong-Ru Chen [1,2], Joseph O. Uweru [1,2], Saipranusha Amancherla[1], Zainab Calcuttawala[1], Antony Brayan Campos-Salazar[1,2], Bruce Corliss [3], Lara Jabbour[1], Jordan Benderoth[1], Bria Friestad[1,2], William A. Mills III[1,2,3], Brant E. Isakson [3,4], Marie-Ève Tremblay [5,6,7,8,9], Chia-Yi Kuan[1,2] & Ukpong B. Eyo [1,2,3✉]

Microglia are brain-resident immune cells with a repertoire of functions in the brain. However, the extent of their interactions with the vasculature and potential regulation of vascular physiology has been insufficiently explored. Here, we document interactions between ramified CX3CR1+ myeloid cell somata and brain capillaries. We confirm that these cells are *bona fide* microglia by molecular, morphological and ultrastructural approaches. Then, we give a detailed spatio-temporal characterization of these capillary-associated microglia (CAMs) comparing them with parenchymal microglia (PCMs) in their morphological activities including during microglial depletion and repopulation. Molecularly, we identify P2RY12 receptors as a regulator of CAM interactions under the control of released purines from pannexin 1 (PANX1) channels. Furthermore, microglial elimination triggered capillary dilation, blood flow increase, and impaired vasodilation that were recapitulated in P2RY12$^{-/-}$ and PANX1$^{-/-}$ mice suggesting purines released through PANX1 channels play important roles in activating microglial P2RY12 receptors to regulate neurovascular structure and function.

[1] Department of Neuroscience, University of Virginia School of Medicine, Charlottesville, VA, USA. [2] Center for Brain Immunology and Glia, University of Virginia, Charlottesville, VA, USA. [3] Robert M. Berne Cardiovascular Research Center, University of Virginia School of Medicine, Charlottesville, VA, USA. [4] Department of Molecular Physiology and Biophysics, University of Virginia School of Medicine, Charlottesville, VA, USA. [5] Axe Neurosciences, Centre de recherche du CHU de Québec—Université Laval, Québec, QC, Canada. [6] Département de médecine moléculaire, Université Laval, Québec, QC, Canada. [7] Department of Neurology and Neurosurgery, McGill University, Montréal, QC, Canada. [8] Division of Medical Sciences, University of Victoria, Victoria, BC, Canada. [9] Biochemistry and Molecular Biology, Faculty of Medicine, The University of British Colombia, Vancouver, BC, Canada. [10]These authors contributed equally: Kanchan Bisht, Kenneth A. Okojie, Kaushik Sharma. ✉email: ube9q@virginia.edu

The brain is an energy-demanding organ[1–3] and its energy demands are met by a rich and dense supply of blood vessels[4]. While peripheral organs allow less restricted entry and exit of substances in the blood circulation, the brain's vasculature is protected structurally by components of the blood–brain barrier (BBB) that allow very restricted entry of blood contents into the brain[5–7]. The brain vasculature is also different from the vasculature of other organs as it consists of the neurovascular unit (NVU), which is made up of various cell types consisting of vascular (i.e., endothelial cells, pericytes, smooth muscle cells, and perivascular macrophages (PVMs)) and brain (i.e., astrocytes and neurons) cells[8,9].

Microglia are the brain's immune cells[10–12] that play important roles in development, mature homeostasis, and disease[12–14]. During development, microglia are known to facilitate optimal vascular complexity development[15–17] and regulate vascular development in various pathologies[18]. For example, microglial elimination in Alzheimer's disease increases vascular hemorrhage and amyloid beta deposition on blood vessels reminiscent of cerebral amyloid angiopathy[19], suggesting protective roles for the vasculature by microglia. In addition, microglia react to systemic inflammation with increased vascular interactions[20] and play a dual role in regulating BBB integrity following an inflammatory insult[21]. Finally, in response to acute vascular injury, microglial processes extend to "plug the leak" and restore vascular integrity[22]. However, the degree to which they interact with the vasculature in homeostasis has not been adequately studied[23,24]. Because the vascular system is essential for the delivery of oxygen and nutrients to (as well as the elimination of waste from) the brain, an understanding of microglial interactions with the vasculature is vital to our understanding of brain homeostasis.

In the current study, using acute and longitudinal in vivo two-photon imaging we identified ramified myeloid cells that stably interact with brain capillaries with their cell bodies. Molecular, morphological, and electron microscopic analyses identified these cells as bona fide microglia that are resident in the brain proper rather than PVMs that reside in the perivascular space. Characterization of capillary-associated microglia (CAMs) revealed (i) that they are enriched on the vasculature (about a third of the microglial population) than is expected at random based of the brain blood vessel density; (ii) that while microglial processes are known to make transient physical contacts with the neurovasculature, these CAM interactions occur through the more stable microglial somata; (iii) that they exhibit mostly similar features with parenchymal microglia (PCMs), suggesting that they are unlikely to be ontogenically or functionally distinct cells; and (iv) that their interactions with capillaries are at least in part regulated by purinergic P2RY12 signaling activated by purines released from pannexin 1 (PANX1) channels. Finally, pharmacological treatment to eliminate microglia resulted in an increase in capillary diameter, cerebral blood flow (CBF), and impairment in vasodilative responses, which were recapitulated with P2RY12 and PANX1 deficiencies. Together, these results provide evidence into the identification, characterization, interaction mechanisms, and functional significance of CAMs, highlighting microglia as bona fide components of the NVU.

## Results

**Ramified CX3CR1$^+$ myeloid cells associate with brain capillaries.** To begin to study brain-resident myeloid cell interactions with the neurovasculature, we performed in vivo two-photon imaging on CX3CR1$^{GFP/+}$ mice. We observed CX3CR1$^+$ cells that had their cell bodies attached to the vasculature. These vessel-associated CX3CR1$^+$ cells were present on blood vessels of all sizes and through the depth of the cortex analyzed (Fig. 1a–d and Supplementary Video 1).

We focused on capillaries (ranging from ~5 to 10 μm in diameter) because the capillary bed represents the most elaborate component of the vasculature, the site of oxygen/nutrient delivery and waste uptake, and often undergo the most elaborate remodeling. Capillaries exhibited closely attached ramified GFP$^+$ cell bodies (Fig. 1e). Close examination revealed that these cells were in the same focal plane as the adjacent capillary and their ramifications extended into the brain proper, suggesting that they are brain-resident (Fig. 1f–h).

Astrocytes are well known to interact with the vasculature as a component of the NVU[25]. To compare the density of these capillary-associated myeloid cell bodies to the cell bodies of astrocytes, we examined capillary-associated astrocyte cell bodies in 1-month-old ALDH1L1$^{GFP/+}$ mice[26]. Capillary-associated myeloid cell body density was significantly greater than capillary-associated astrocyte cell body density on capillaries in the cortex in vivo (Fig. 1i–j). However, capillary-associated myeloid cell body density was much lower than capillary-associated pericyte (which reside in the perivascular space) cell body density[27]. Taken together, these findings indicate that ramified CX3CR1$^+$ myeloid cell bodies associate with brain capillaries.

**Ramified CX3CR1$^+$ capillary-associated myeloid cells are bona fide microglia.** Brain myeloid cells include ramified microglia, PVMs, meningeal macrophages, and choroid plexus macrophages[28]. These cortical capillary-associated GFP$^+$ cells were ruled out to be meningeal or choroid plexus macrophages. However, PVMs are localized between the two vascular basement membranes, and microglia are localized outside the basement membrane of the vascular wall in the brain parenchyma. Electron microscopy identified myeloid cells associated with capillaries in the brain parenchyma (outside the basement membranes) in different brain regions (Fig. 2a) consistent with the previous reports[21,23,24,29]. Furthermore, we noted that the cell bodies of these myeloid cells were directly adjacent to the corresponding vessel and contacted the basement membrane without intervening astrocytic processes as has recently been reported elsewhere[24].

CX3CR1$^{GFP/+}$ tissues stained with CD206- (a PVM marker[28]) labeled cells in the meninges and cortical parenchyma (Supplementary Fig. 1a–d and Table 1). Remarkably, we observed that most GFP$^+$ cells were CD206-negative (CD206$^-$) and most CD206-positive (CD206$^+$) cells in the brain were localized to large blood vessels and the meninges (Fig. 2b, d and Supplementary Fig. 1a–d). Of all labeled cells, 90.8 ± 1.45% were GFP$^+$ only, 6.03 ± 1.45% were CD206$^+$ only, and 3.19 ± 0.04% were double positive for GFP and CD206. CX3CR1$^{GFP/+}$ tissues were then stained with P2YR12 (a microglial-specific marker[28,30–32]). Occasionally, GFP$^+$;P2RY12$^-$ cells could be detected in larger blood vessels (arrowheads in Supplementary Fig. 1e–h and Supplementary Fig. 2a–e and Table 1). Typically, these cells lacked ramified processes (unlike P2RY12$^+$ cells) and sometimes expressed lower GFP levels (arrowhead in Supplementary Fig. 2a–c and right arrowhead in Supplementary Fig. 2d–e and Table 1). Virtually all non-meningeal GFP$^+$ cells were P2RY12$^+$ (97.16 ± 0.52%; Fig. 2c, e), and a population of the cell bodies of these cells was often localized to capillaries (arrows in Supplementary Fig. 3a–c). Therefore, we refer to these capillary-interacting, ramified CX3CR1$^+$ (i.e., GFP$^+$) cells as CAMs.

CAMs were present at similar densities in the cerebral cortex, thalamus, and hippocampus (Fig. 2f). Carefully generated 3D reconstructions of CX3CR1$^+$ cell somata and the vasculature revealed physical interactions between the two brain elements (Fig. 2g and Supplementary Video 2). The vasculature occupied 6.1 ± 0.93% of the total brain volume, whereas ramified CAMs

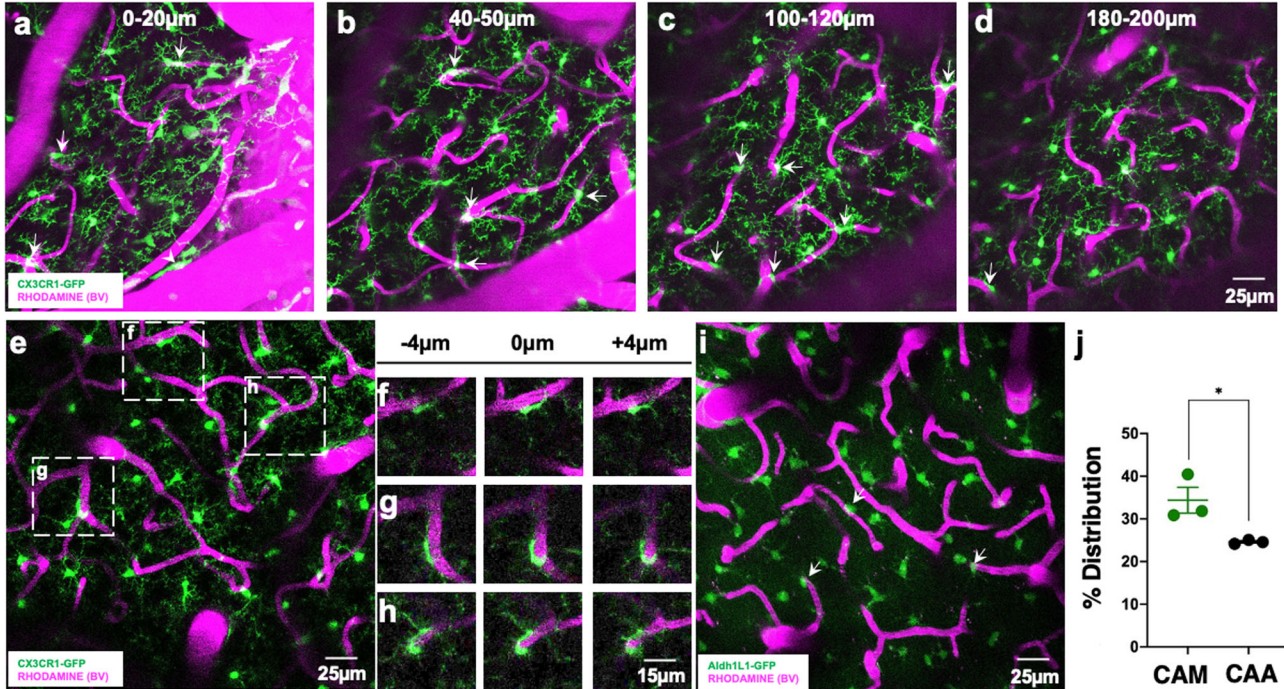

**Fig. 1 Ramified CX3CR1+ myeloid cells associate with brain capillaries. a–d** Representative 20-μm-thick two-photon projection images from a CX3CR1GFP/+ adult brain showing myeloid cells (green) and the vasculature (rhodamine in magenta) at varying tissue depths between the brain surface and 200 μm of the cortex. Arrows identify capillary-associated ramified myeloid cells. **e** Representative 20-μm-thick in vivo two-photon image from a CX3CR1GFP/+ adult brain showing ramified myeloid cells (green) and the vasculature (rhodamine in magenta) in the cortex. **f–h** Representative images from boxed regions in (**e**) showing somal interactions between the ramified myeloid cells and capillaries in an 8 μm tissue volume. **i, j** Representative 20-μm-thick two-photon projection image (**i**) and quantification (**j**) from an ALDH1L1GFP/+ P30 brain showing astrocytes (green) and the vasculature (rhodamine in magenta) in the cortex. Capillary-associated astrocytes (CAAs, arrows in **i**) density is compared to capillary-associated myeloid (CAM) density. $n = 3$ mice each. Representative images in (**a–h**) were observed in five mice and in (**i**) was observed in three mice. Data are presented as mean values ± SEM. *$p < 0.05$. Two-sided unpaired Student's $t$ test.

accounted for ~30% of the microglial population across brain regions (Fig. 2h). We reasoned that if 6% of the brain's volume is occupied by blood vessels, then we could expect ~6% of microglial cell bodies to associate with the vasculature of the total microglial population if placed randomly in the brain. Our finding of ~30% of microglial cell bodies associating with capillaries in the brain, therefore, indicates that CAMs are 5× more enriched on the vasculature above what would be expected at random. Taken together, these results indicate that ramified CX3CR1+ myeloid cells are bona fide microglia.

**Comparison between CAMs and PCMs.** Based on their position along the capillary wall, at least three CAM categories were identified including *lined* CAMs whose somata are aligned parallel to the corresponding capillary; *wrapped* CAMs with bipolar processes that wrap around the capillary; and *junctional* CAMs whose somata reside at capillary bifurcations (Supplementary Fig. 4a). CAM density was maintained between P15 and 12 months at ~30% (Supplementary Fig. 4b), and they could be detected in the neonatal (P5) brain (Supplementary Fig. 4c and Supplementary Video 3). Male and female adult mice showed similar blood vessel volume and CAM density (Supplementary Fig. 4d). Interestingly, CAM density was increased with chronic window implantation when compared to an acute window preparation, suggesting that the window implantation approach slightly but significantly increases CAM interactions (Supplementary Fig. 4e).

CAMs are distinguished from PCMs by their position. Therefore, to further characterize CAMs, we compared the expression of *Sall1*, a unique microglial transcription factor

among myeloid cells[33] and noted no significant differences (Fig. 3a, b and Table 1). To interrogate possible static morphological and dynamic functional differences between CAMs and PCMs, we compared their static and dynamic morphological features. CAMs showed a slight but significant reduction in primary process numbers (Fig. 3c, d), a larger cell body area (Fig. 3e), but an identical cell territory (Fig. 3f). In response to a laser-induced injury, both CAMs and PCMs exhibited directed process extension (Fig. 3g–j and Supplementary Video 4). Longitudinal imaging of microglia daily or weekly (Fig. 3k), identified stable, crawling, "hop on" and "hop off" CAMs (Fig. 3l–o). When monitored over a month, stable CAMs were the most abundant CAM category (Fig. 3m, p). Together, these results suggest that CAMs represent an interchangeable microglial population with PCMs with extended cell body residence on capillaries.

**P2RY12 and PANX1 channels regulate CAM interactions.** A small population of microglia (~5–8%) in the cerebral cortex migrate daily under purinergic P2RY12 control[34]. Therefore, we investigated P2RY12 roles in CAM interactions and dynamics. First, we found no gross defects in vascular density in P2RY12−/− mice (6.1 ± 0.93% in wild-type and 5.5 ± 0.43% in P2RY12−/− mice; Fig. 4a). However, both male and female P2RY12−/− mice showed a significantly reduced CAM density (Fig. 4b–d). One allele of P2RY12 was sufficient to ensure normal CAM density (Fig. 4e). The reduction in CAM density was not a result of a reduced number of microglia in P2RY12−/− mice because these mice had a slight but significantly greater microglial density (Fig. 4f). The average spacing between CAMs along the

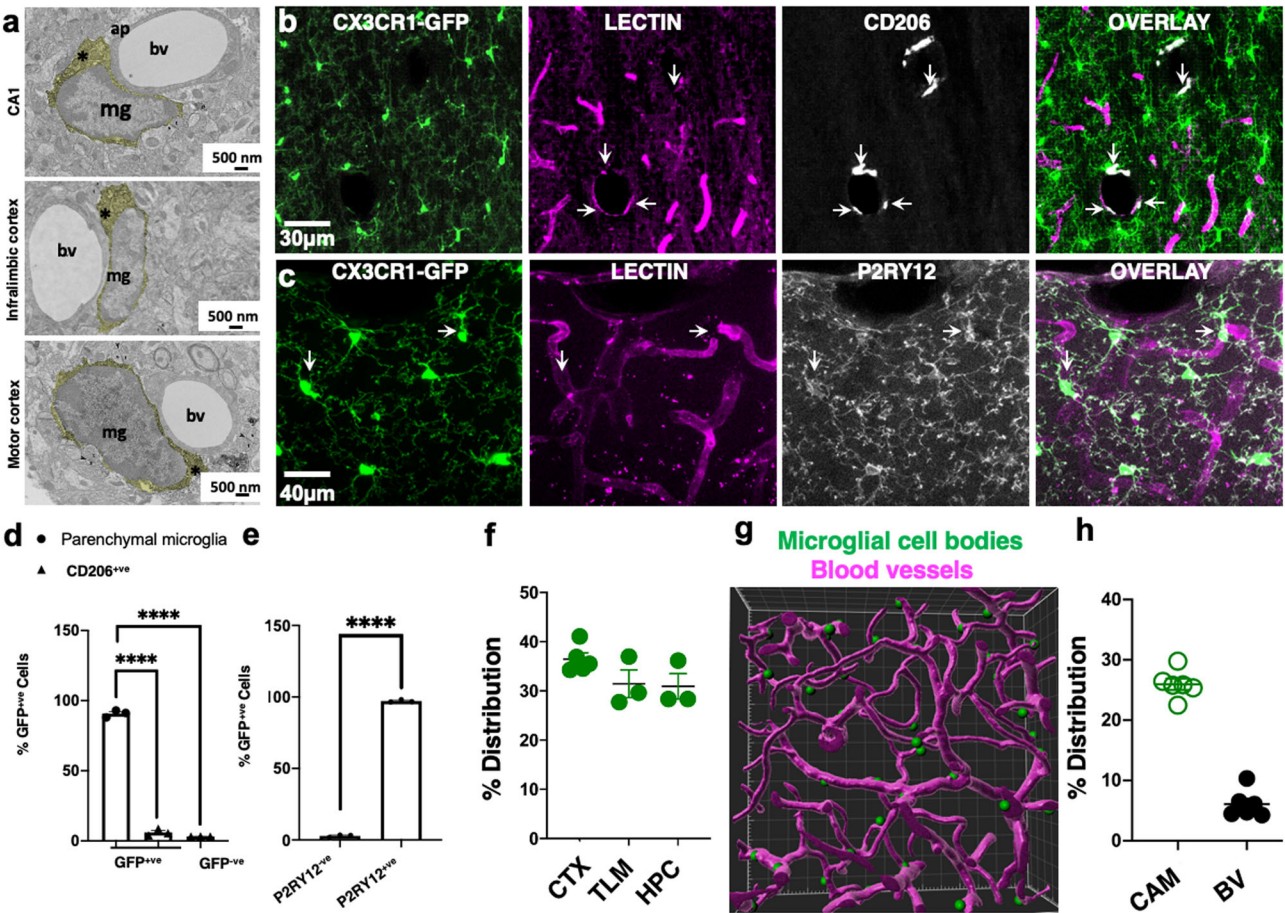

**Fig. 2 Ramified CX3CR1⁺ capillary-associated myeloid cells are bona fide microglia. a** Electron microscopy images of capillary-associated microglia (mg) showing their cytoplasm (*), directly adjacent to blood vessels (bv) and astrocytic process (ap) in various brain regions. Microglial cell cytoplasm is psuedocolored yellow. Representative images were observed in three mice. **b**, **c** Representative projection images from a CX3CR1$^{GFP/+}$ brain showing myeloid cells (green), the vasculature (lectin in magenta) and CD206⁺ (**b**) or P2RY12 (**c**) cells (white). **d**, **e** Quantification of CD206⁺ and CD206⁻ cells (**d**, $n = 3$ mice) or P2RY12⁺ and P2RY12⁻ cells (**e**, $n = 3$ mice). **f** Distribution of capillary-associated microglia in the cortex (CTX), thalamus (TLM), and hippocampus (HPC) as quantified from fixed brain tissues. $n = 3$–5 mice. **g** IMARIS generated 3D reconstructed image from two-photon data of microglial somata (green) associated with the vasculature (magenta). **h** Distribution of capillary-associated microglia (CAMs) amongst total microglia and blood vessel (BV) volume in total brain volume. $n = 6$ mice. Data are presented as mean values ± SEM. ****$p < 0.0001$. Two-sided unpaired Student's $t$ test in (**e**).

vasculature was also greater in P2RY12$^{-/-}$ mice ($387 \pm 64\,\mu m$) compared to wild-type mice ($223 \pm 31\,\mu m$; Fig. 4g), although not statistically significant with the number of mice assessed, suggesting possibly poorer vascular surveillance in P2RY12$^{-/-}$ mice. Finally, monitoring CAM dynamics for a month revealed no statistical differences between stable and dynamic CAM subtypes in both genotypes (Fig. 4h), indicating that such dynamic interactions are not regulated by P2RY12.

Since P2RY12 is activated by purines[35], which can be released from PANX1 channels[36], we examined microglial density and CAM density in PANX1$^{-/-}$ mice and determined (as with a P2RY12 deficiency), that microglial density was significantly increased (Fig. 4i, j) and CAM interactions were significantly reduced in PANX1$^{-/-}$ mice (Fig. 4i, k). Together, these results indicate that P2RY12- and PANX1-dependent mechanisms regulate microglial density and CAM interactions.

**CAM interactions are not altered by increased neuronal activity.** Next, we asked whether CAMs are influenced by neuronal activity. To this end, we chronically implanted cranial windows and followed CAMs after kainic acid (KA)-induced seizures. We have previously shown that following severe seizures (i.e., stage 5 seizure scores along a modified Racine scale) induced

by systemic KA treatment, microglia show increased cortical positional rearrangement of their landscape by translocation[34]. However, we did not assess the consequence of this approach to increase neuronal activity in vivo on CAMs. Therefore, we monitored CAMs daily before and up to 48 h after KA-induced seizures. Despite some changes in the microglial morphology from smaller somata to larger ones during this period, the total microglial density remained unchanged (Fig. 5b) as did the percent of the CAM population (Fig. 5c). Similarly, CAM dynamics following seizures (Fig. 5d) was identical to that in the basal condition (Fig. 3p). Moreover, this was the same when hippocampal microglial (Fig. 5e) and CAMs (Fig. 5f) density was assessed in fixed tissues following seizures, suggesting that CAM interactions and dynamics are not altered by increased neuronal activity. Importantly, KA treatment did not induce an increase in BBB permeability as assessed by Evan's Blue extravasation (Supplementary Fig. 5).

**Microglia regulate the vascular structure.** To determine whether CAMs are an intrinsic or random subset of microglia, we eliminated and replenished microglia using PLX3397, a CSF1R inhibitor[37]. Notably, both CAMs and PCMs expressed similar levels of CSF1R (Supplementary Fig. 6 and Table 1), indicating

**Table 1 Immunostaining conditions.**

| | Quenching | Blocking | Primary | Secondary |
|---|---|---|---|---|
| IBA1 | 2% $H_2O_2$ in 70% methanol for 10 min | 10% FCS + 3% BSA + 0.5% Triton, in Tris-buffered saline for 1h at room temperature | 1:800 in blocking buffer O/N at 4°C | 1:500 anti-rabbit antibody in Tris-buffered saline with 0.5% Triton |
| CSF1R | 0.3% $H_2O_2$ for 10 min 0.1% $NaBH_4$ for 30 min | 10% NGS + 0.2% Triton X-100 in PBS at room temperature for 1h | 1:250 in blocking buffer O/N at 4°C | 1:500 goat anti-rabbit in PBS + 0.2% Triton X-100 for 1.5h at room temperature |
| CD13 | - | 1% BSA + 2% Triton in phosphate-buffered saline for 1h at room temperature | 0.8 µg/mL in blocking buffer O/N at 4°C | 1:500 anti-goat in phosphate-buffered saline |
| AQP4 | - | 10% NGS + 0.1% Triton in phosphate-buffered saline for 1h at room temperature | 1:400 in blocking buffer O/N at 4°C | 1:500 anti-rabbit in phosphate-buffered saline |
| CD206 | 2% $H_2O_2$ in 70% methanol for 10 min | 10% FCS + 3% BSA + 0.05% Triton, in Tris-buffered saline for 1h at room temperature | 1:300 in blocking buffer O/N at 4°C | 1:500 anti-rat antibody in Tris-buffered saline with 0.05% Triton |
| P2RY12 | - | 10% NGS + 0.4% Triton, in Tris-buffered saline for 1h at room temperature | 1:400 in blocking buffer O/N at 4°C | 1:500 anti-rabbit antibody in Tris-buffered saline |
| CD31 | - | 5% BSA + 0.3% Triton in phosphate-buffered saline for 1h at room temperature | 1:150 in blocking buffer O/N at 4°C | 1:500 anti-Armenian hamster in phosphate-buffered saline with 0.3% Triton |

that both microglial groups can be expected to be targeted by PLX3397. Treatment with PLX3397 effectively eliminated microglia (and does not affect BBB permeability[37]), while withdrawal of drug treatment resulted in its rapid repopulation (Fig. 6a, b). We observed in vivo what has been previously described in fixed brain slices[37], i.e., that repopulating microglia show larger cells bodies and shorter processes (Fig. 6a). During PLX3397 treatment, total microglia and CAMs decreased and recovered after PLX3397 withdrawal. However, the CAM density of the residual microglial population remained constant (~30% of residual microglia) throughout the elimination–repopulation cycle (Fig. 6b), suggesting that CAM density in the microglial landscape is intrinsically regulated.

Microglial elimination during PLX3397 treatment preserved the overall vascular structure, i.e., no new vessels were formed, and no extant vessels were lost. However, while capillary diameter was maintained under controlled conditions (Fig. 6c–f, k), capillary diameter after 4 days of PLX3397 treatment increased by ~15% (Fig. 6g–k), suggesting that microglia contribute to the maintenance of capillary diameter.

**Microglia regulate vascular function through PANX1–P2RY12 coupling.** In addition, as a consequence of dilated capillaries, we measured CBF using laser-speckle imaging and detected a ~20% increase with PLX3397 treatment (Fig. 7a–d). Furthermore, with PLX3397 treatment, response to $CO_2$ (a vasodilative agent) showed a trend towards impairment (Fig. 7e–f), suggesting that microglia regulate both blood flow dynamics and are necessary for at least some aspects of vascular reactivity. Given our observation that both P2RY12- and PANX1- deficient mice show reduced CAM interactions (Fig. 4), we hypothesized that this molecular mechanism may also regulate CBF and vasodilation. To test this hypothesis, we repeated laser-speckle imaging in age-matched wildtype, P2RY12$^{-/-}$ and PANX1$^{-/-}$ mice. Consistent with the PLX3397 findings, we observed increased basal perfusion levels as well as impaired vasodilation to $CO_2$ with both a P2RY12 (Fig. 7g–i) and PANX1 deficiency (Fig. 7j–l). Because pericytes and astrocytes also regulate vascular function, we assessed pericyte cell density and astrocytic endfeet density in PLX3397-treated and P2RY12$^{-/-}$ mice. Neither the depletion of microglia nor a P2RY12 deficiency altered pericyte cell density or astrocytic endfeet density (Supplementary Fig. 7 and Table 1). Together, these results suggest that purines released from PANX1 facilitate microglial–capillary interactions, which function to regulate cerebrovascular perfusion and reactivity in the steady state.

## Discussion

In this study, we have characterized the enrichment of microglia on capillaries and provided evidence for microglial involvement in the maintenance of capillary structure and regulation of CBF and vasodilation. Specifically, we show (i) that capillary-associated CX3CR1-expressing cells are *bona fide* microglia (or CAMs) using ultrastructural, morphological and molecular approaches; (ii) that CAM interactions are usually stably confined to capillaries but they sometimes can be interchanged with PCMs; (iii) that CAM interactions are regulated by a molecular mechanism that involves microglial-specific P2RY12 which are activated by purines released from PANX1 channels; (iv) that CAM density seems to be governed by a brain-intrinsic program to maintain a third of the microglial population on capillaries irrespective of the overall microglial pool; and (v) that microglia contribute to the maintenance of optimal capillary diameter, CBF and vascular responsiveness through the PANX1–P2RY12 coupling in the healthy state (Fig. 8). To our knowledge, this is the first report showing microglial contributions to capillary structure

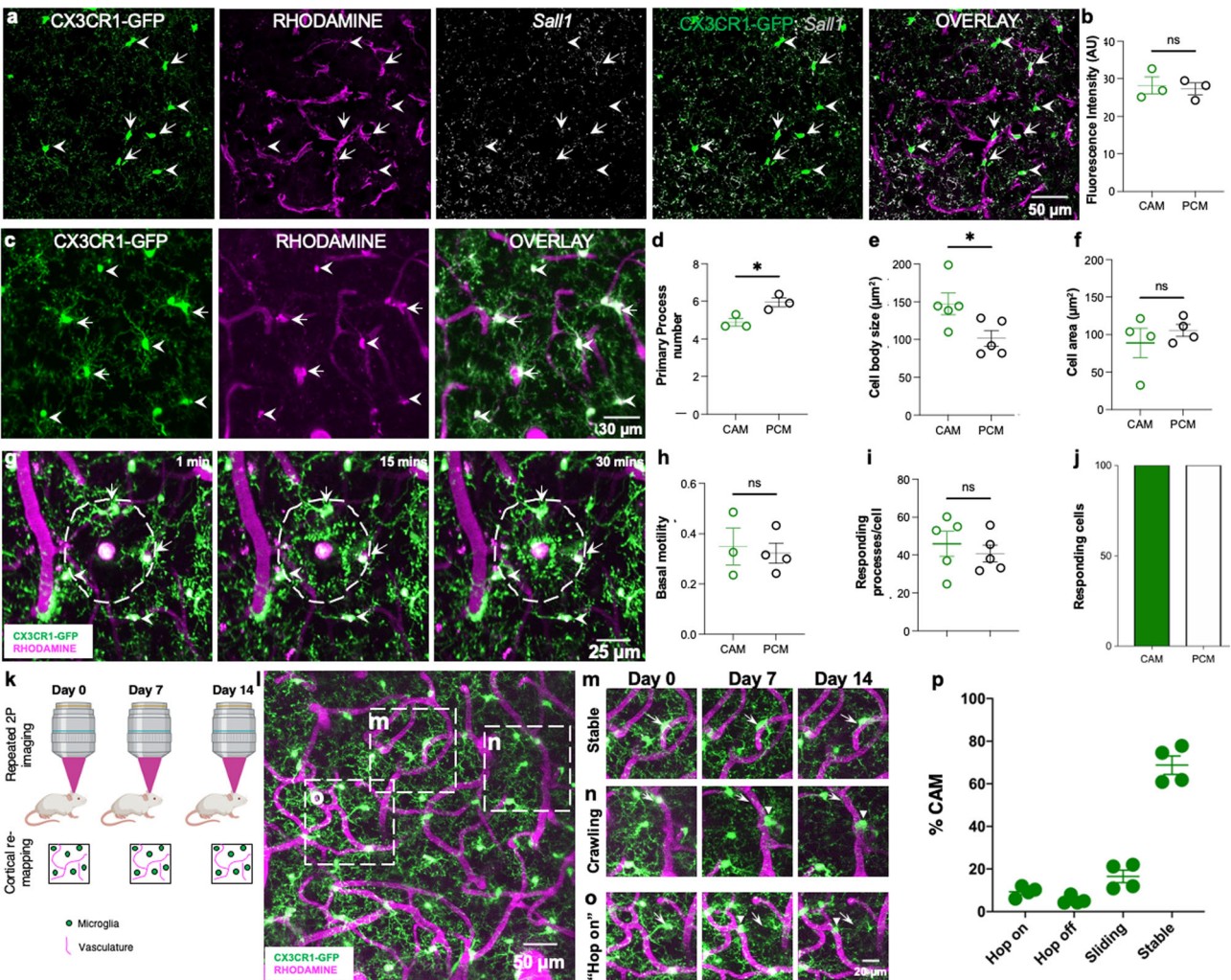

**Fig. 3 Comparison between capillary-associated microglia and parenchymal microglia. a** Representative images from a CX3CR1[GFP/+] adult brain showing microglia (green) capillaries (magenta) and *Sall1* transcripts (gray) with capillary-associated microglia (CAMs, arrows) parenchymal microglia (PCMs, arrowheads). **b** Quantification of microglial *Sall1* expression in CAMs and PCMs. n = 10 CAM or PCM cells from each of three mice. **c** Representative in vivo two-photon projection images from a CX3CR1[GFP/+] adult brain showing microglia (green) that are either capillary- (magenta) associated (arrows) or parenchyma-situated (arrowheads). **d–f** Quantification of microglial primary process numbers (**d**, n = 5 CAM or PCM cells from each of five fields of view from each of three mice), microglial cell body sizes and (**e**, n = 5 CAM or PCM cells from each of five fields of view from each of five mice) microglial whole-cell area between CAMs and PCMs (**f**, n = 5 CAM or PCM cells from each of three to five fields of view from each of four mice). **g** Representative in vivo two-photon projection images from a time-lapse movie collected from a CX3CR1[GFP/+] mouse following a laser-induced injury. Microglial processes are directed towards the laser injury over time. **h–j** Quantification of basal microglial motility (**h**, n = 3–4 CAM or PCM cells from each of three mice) number of responding processes per cell (**i**, n = 3–4 CAM or PCM cells each of five mice) and the percent of responding cells following the laser-induced injury between CAMs and PCMs (**j**). **k** A representative schematic showing the longitudinal imaging scheme for visualizing microglia and capillaries. **l–o** Representative two-photon projection image from a CX3CR1[GFP/+] adult brain during longitudinal imaging showing various microglial (green)–capillary (magenta) dynamics including stable interactions, which persist over the imaging period (**m**), crawling interactions in which the position on the capillary changes, but remains on the capillary (**n**) and "hop on," in which parenchymal microglia relocate to a proximal capillary (**o**). **p** Quantification of the various types of dynamic CAMs over a 4-week imaging period. n = 3–5 fields of view from each of four mice. Data are presented as mean values ± SEM. *p < 0.05, and n.s. not significant. Two-sided unpaired Student's t test.

and function outside pathology. These findings have significant implications as a foundation to understand microglial contributions to vascular function that may be relevant especially in microvascular disease.

There has been growing interest in microglial diversity in the brain[38,39], which has been revealed in recent years by transcriptional differences between microglia across and within species in health[40,41] and following injury[42,43]. In addition to transcriptional differences, microglial diversity has been shown with regard to functional status. For example, with increased diversity in development, a subset of microglia have been identified as proliferation-associated microglia[44]. Furthermore, disease-associated microglia in neurodegeneration have recently been described[45]. In addition, "subsets" of microglia have been identified based on their location without known transcriptional differences. For example, axon initial segment (AXIS) microglia were discovered in relation to their position along the neuronal axon initial segment[46] and vessel-associated microglia (VAMs) were recently described in relation to their position along blood vessels, although the identity of the blood vessels (capillaries, arterioles, veins, arteries) were not clarified[21].

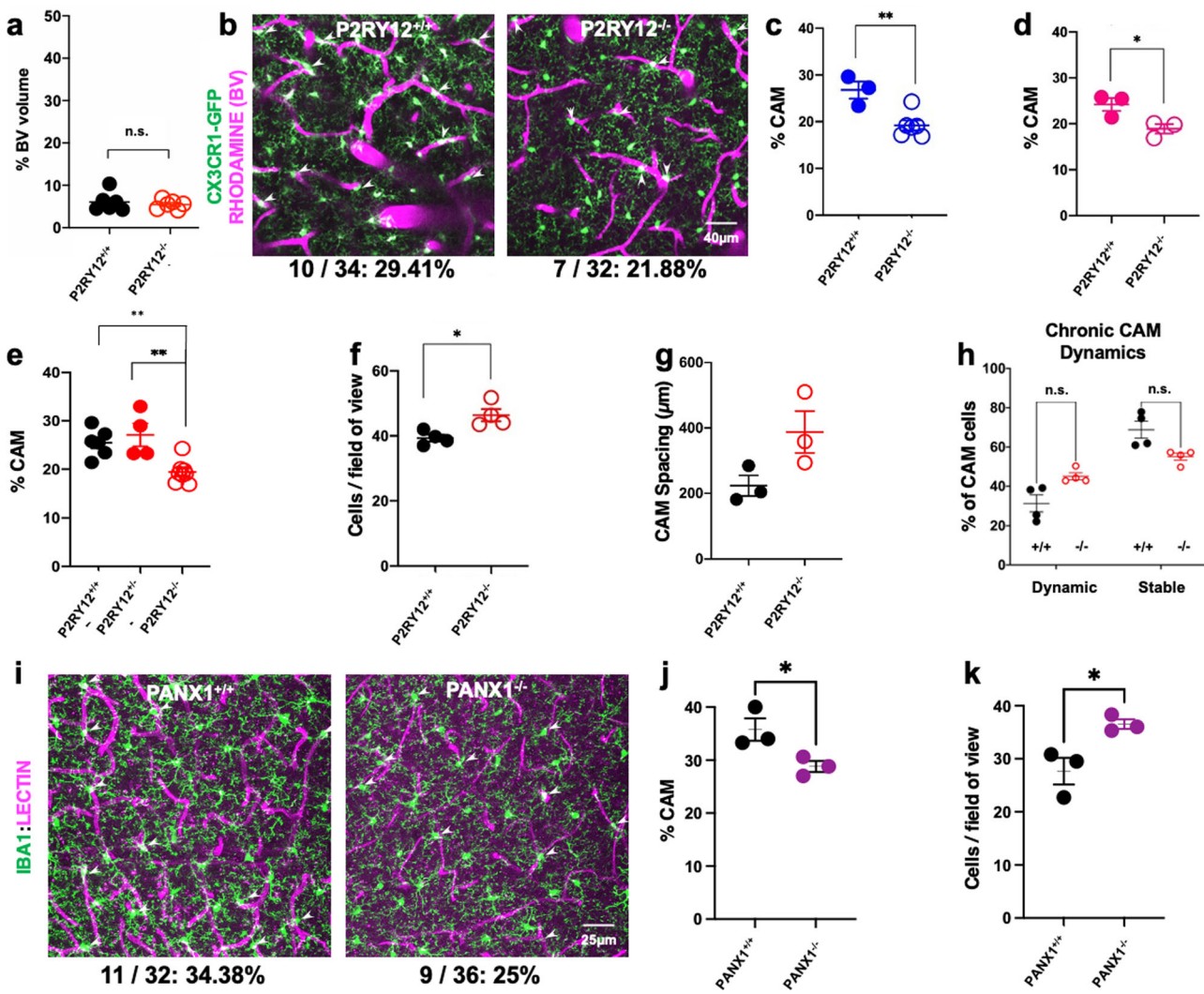

**Fig. 4 P2RY12- and PANX1-dependent regulation of CAM interactions. a** Quantification of the blood vessel volume when compared to total brain volume in P2RY12[+/+] and P2RY12[−/−] mice. $n = 6$ mice each. **b–d** Representative 20-μm-thick two-photon projection images (p) and quantification in males (**c**, $n = 3$–5 mice each) and females (**d**, $n = 3$ mice each) from a CX3CR1[GFP/+] adult brain showing capillary- (magenta) associated microglia (green, arrowheads) in P2RY12[+/+] and P2RY12[−/−] mice ($n = 3$–5 mice each). **e** Quantification of CAM density relative to total microglial density in P2RY12[+/+], P2RY12[+/−], and P2RY12[−/−] mice ($n = 4$–6 mice each). **f** Quantification of microglial density in the in vivo cortical regions imaged in P2RY12[+/+] and P2RY12[−/−] mice ($n = 4$ mice each). **g** Quantification of the spacing between microglia along the vasculature in P2RY12[+/+] and P2RY12[−/−] mice ($n = 3$ mice each). **h** Quantification of CAM dynamics in P2RY12[+/+] and P2RY12[−/−] mice ($n = 4$ mice each). **i–k** Representative 20-μm-thick confocal projection images (**i**) and quantification of CAM density (**j**) and microglial cell density- (**k**) showing capillary- (magenta) associated microglia (green, arrowheads) in PANX1[+/+] and PANX1[−/−] mice ($n = 3$ mice each). Data are presented as mean values ± SEM. *$p < 0.05$, **$p < 0.01$, and n.s. not significant. Two-sided unpaired Student's $t$ test in **a**, **c**, **d**, **f**, **g**, **h**, **j**, and **k**.

Others have previously[47] and recently[24] described "juxtavascular microglia" along blood vessels. In this context, "juxtavascular" referred to the association of the microglial cell body or process with the vasculature. "Juxtavascular" microglial cell bodies could be situated up to 50 μm from blood vessels, which varied in size from 4–30 μm[47]. This was examined in the neonatal (P4–10) rat hippocampus leaving open the question of microglia–vascular interactions in maturity. A more recent study using the same nomenclature identified juxtavascular microglia as increased in the early postnatal (P1-5) mouse and embryonic human brain that were reduced with maturity. This study showed that these interactions were predominantly directed to capillaries where CX3CR1 was shown to regulate their colonization of the brain. In maturity, these cells were stabilized at capillaries[24], which we also found in the current study.

For our interests, we have focused our attention on microglial cell body interactions at microvascular capillaries (5–10 μm in diameter) because these are more stable than the highly dynamic microglial processes[48,49]. We confirmed the existence of microglial cell bodies associated with capillaries, which we call CAMs for precision. It is likely that CAMs are (at least a subset) of previously described VAMs[21] or juxtavascular microglia[24,47]. We report that CAMs are highly enriched on capillaries at a rate that is 5× greater than would be expected at random. They make up about a third of the microglial population, a number that is similar to previous reports of VAMs or "juxtavascular microglia"[21,24,47]. Molecular approaches identified CAMs as expressing microglial-specific P2RY12 and similar levels of *Sall1* transcripts and CSF1R protein to PCMs. Moreover, like PCMs that are not associated with capillaries, CAMs are ramified, respond chemotactically to a laser-induced injury, and can be

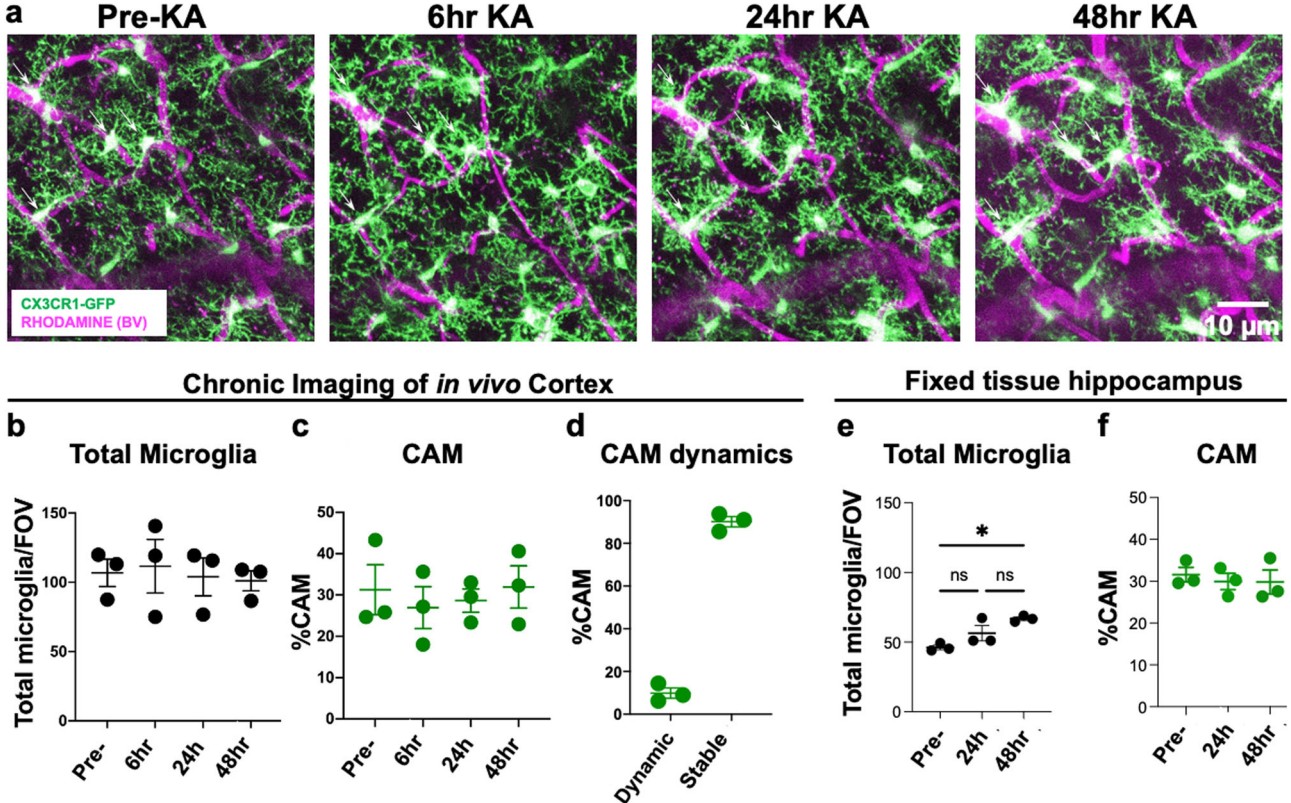

**Fig. 5 CAM interactions are not altered by increased neuronal activity. a** Representative in vivo two-photon projection images from the same field of view in a CX3CR1$^{GFP/+}$ adult brain showing capillary- (magenta) associated microglia (green) before and up to 48 h after KA-induced seizures. **b**, **c** Quantification of total microglial (**b**) and capillary-associated microglial (**c**) density over time before and following seizures. **d** Quantification of the stable and dynamic CAMs following seizures. **e**–**f** Quantification of total microglial (**e**) and capillary-associated microglial (**f**) density from fixed slices in the hippocampus following seizures. $n = 3$ mice each. Data are presented as mean values ± SEM. *$p < 0.05$. n.s. not significant. Two-sided unpaired Student's $t$ test in **b**, **c**, **e**, and **f**.

depleted with CSF1R inhibition using PLX3397. Remarkably, even during PLX3397-induced microglial depletion or repopulation, CAMs maintain ~30% of the total microglial population, suggesting an intrinsic brain programming that ensures that a third of the microglial population remains on capillaries. Since microglial repopulation following PLX3397 treatment occurs solely from microglial proliferation without contributions from the periphery[50], these findings suggest a brain-intrinsic programming of CAM interactions. The molecular basis for this intrinsic regulation remains unknown, but perhaps the vasculature could be a source of prosurvival microglial factors such as CSF1 and/or IL-34 known to regulate microglial survival and maintenance[51].

Having clarified CAM identity, we set out to understand mechanisms regulating CAM interactions. Mechanisms regulating CAM interactions could be broadly classified into two types: (i) short-term attraction mechanisms and (ii) long-term attachment mechanisms. Given that microglia are well known to express purinergic receptors including P2RY12, which regulate microglial process[35,52,53] and cell body movements[34] and PANX1 channels present on the vasculature can release purines[54–56], we tested the roles of P2RY12 and PANX1 in regulating CAM density. We consistently found a reduction of CAM density in respective knockout mice for these proteins suggesting a PANX1–P2RY12 regulating the attraction of microglia to capillaries. While this mechanism could serve as an attraction mechanism, purines are rapidly degraded in the extracellular space by enzymes and are unlikely to serve as a long-term attachment mechanism for CAM interactions, which could

presumably be regulated by other yet-to-be-identified adhesion factors.

In addition to identifying CAMs and some of the molecular mechanisms regulating their interactions, using pharmacological and genetic approaches, we assessed the consequence of a loss of microglia and CAM-interaction mechanisms on capillary structure and function. Without altering pericyte or astrocytic endfeet density, microglial elimination with PLX3397 resulted in a ~15% increase in capillary size that corresponded with a ~20% increase in cerebrovascular perfusion, but an ~50% reduction in vascular reactivity to a vasodilative $CO_2$ treatment. Some of these findings were recapitulated in P2RY12$^{-/-}$ and PANX1$^{-/-}$ mice, indicating an overlap in the molecular mechanism regulating CAM interactions, cerebrovascular perfusion, and vasodilative reactivity. It remains to be determined whether microglia elicit these effects on the vasculature directly or indirectly via the modulation of an intermediate cell type like astrocytes, pericytes, or neurons that are already known to regulate the vasculature. Moreover, whether microglia play significant roles in vascular reactivity other than $CO_2$-mediated vasodilation remains to be investigated.

In summary, we have clarified the identity of CAMs, elucidated one underlying mechanism regulating these interactions in the PANX1–P2RY12 coupling and provided support that a feature of CAM interactions includes regulation of capillary structure and function. Together, these findings provide strong support for the inclusion of microglia as a significant component of the NVU as well as recommend further studies into microglial contributions to vascular structure and function.

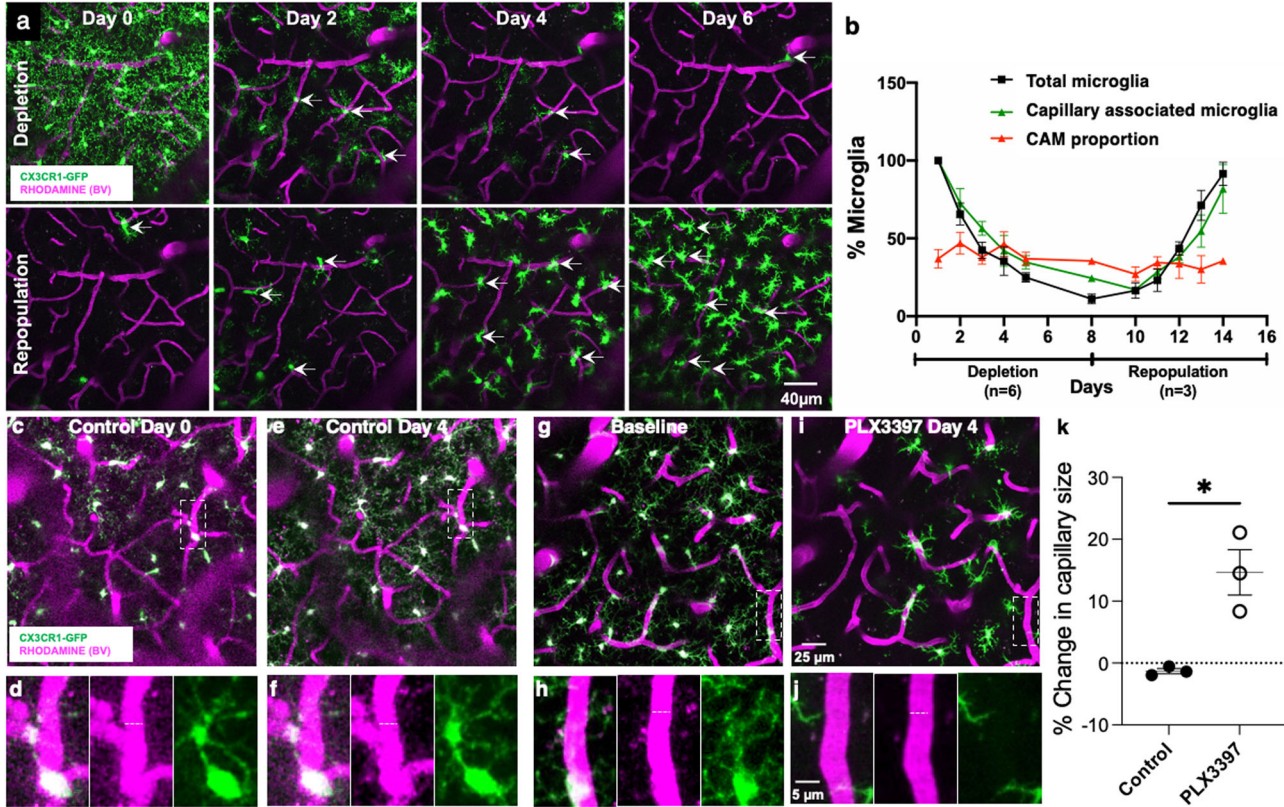

**Fig. 6 Microglia regulate capillary diameter. a** Representative 20-μm-thick in vivo two-photon projection image from a CX3CR1$^{GFP/+}$ adult brain during longitudinal imaging with PLX3397 treatment (top) and withdrawal (bottom) showing microglial (green) depletion and repopulation with the preservation of the vasculature (magenta). **b** Quantification through the time of the depletion and repopulation scheme for all microglia and capillary-associated microglia as well as the proportion of CAMs relative to remaining microglia. $n = 4$ mice. **c**–**f** Representative two-photon projection image from a CX3CR1$^{GFP/+}$ adult brain during longitudinal imaging with a mouse fed control chow. Representative microglia (green) and blood vessels (magenta) in boxed regions in (**c**, **e**) are magnified in (**d**, **f**). Dashed lines indicate the capillary diameter. (**g**–**j**) Representative two-photon projection image from a CX3CR1$^{GFP/+}$ adult brain during longitudinal imaging with a mouse fed PLX3397 chow. A representative microglia in the boxed regions in (**g**) is magnified in (**h**) and depleted in (**i**, **j**). Dashed lines indicate the capillary diameter. **k** Quantification of percent change in capillary size following control or PLX3397 treatment. $n = 3$–7 capillaries from 11 to 13 fields of view from each of three mice. Data are presented as mean values ± SEM. *$p < 0.05$; two-sided unpaired Student's $t$ test.

## Methods

**Animals and treatment**. Animal experiments were conducted in accordance with the relevant guidelines and regulations established and approved by the Institutional Animal Care and Use Committee at the University of Virginia. All animals were housed under controlled temperature, humidity, and light (12:12 h light–dark cycle) with food and water readily available ad libitum. Both male and female mice were used for this study. Heterozygous GFP reporter mice expressing GFP under control of the fractalkine receptor promoter (CX3CR1$^{GFP/+}$)[57] were used for the imaging studies as well as for immunohistochemical staining. For investigating the involvement of purinergic signaling, P2RY12$^{-/-}$:CX3CR1$^{GFP/+}$ mice were generated by crossing P2RY12$^{-/-}$ mice[58] donated by Dr. Long-Jun Wu at Mayo Clinic with homozygous (CX3CR1$^{GFP/GFP}$) mice. The double heterozygote F1 generation was then subsequently re-crossed to generate P2RY12$^{-/-}$:CX3CR1$^{GFP/+}$ mice. PANX1$^{+/+}$ and global PANX1$^{-/-}$ mice[59] were provided by Dr. Brant Isakson. ALDH1L1$^{GFP/+}$ mice[26] were a gift from Dr. Manoj Patel at the University of Virginia.

For microglial depletion studies, mice were fed with chow containing a final dose of 660 mg/kg PLX3397, a CSF1R inhibitor widely used to eliminate microglia from the brain[37]. For microglial repopulation studies, the mice were switched from Plexxikon chow (formulated with Research Diets, New Jersey) back to control chow to allow for microglial repopulation[50].

**Acute and chronic window implantation**. Mice were implanted with a chronic cranial window as previously described[60]. Briefly, during surgery, mice were anesthetized with isoflurane (5% for induction; 1–2% for maintenance) and placed on a heating pad. Using a dental drill, a circular craniotomy of >3 mm diameter was drilled at 2 mm posterior and 1.5 mm lateral to bregma; the craniotomy center was around the limb/trunk region of the somatosensory cortex. A 70% ethanol-sterilized 3 mm glass coverslip was placed inside the craniotomy. A light-curing dental cement (Tetric EvoFlow) was applied and cured with a Kerr Demi Ultra

LED Curing Light (DentalHealth Products). iBond Total Etch glue (Heraeus) was applied to the rest of the skull, except for the region with the window. This was also cured with the LED Curing Light. The light-curing dental glue was used to attach a custom-made head bar onto the other side of the skull from which the craniotomy was performed. For acute imaging, imaging windows were monitored immediately after the surgery. For chronic imaging, mice were allowed to recover from anesthesia for 10 min on a heating pad before returning to their home cage. Mice were allowed to recover from the cranial window surgery for 2–4 weeks before the commencement of chronic imaging. Only surviving mice with a clear glass window were used for the imaging studies.

**In vivo two-photon imaging**. For acute or chronic imaging, mice were anesthetized with isoflurane (5% for induction; 1–2% for maintenance). The head of the anesthetized mice was stabilized and mounted by the head plate and the animal was placed on a heating plate at ~35 °C under the two-photon microscope. One hundred microliters of Rhodamine B dye (2 mg/mL) was injected intraperitoneally or subcutaneously to label the vasculature. For longitudinal imaging, the blood vessel architecture visible through the craniotomy window was carefully recorded as a precise map of the brain region being visualized and was used to trace back to the original imaging site for chronic imaging studies[60]. Imaging was conducted using a Leica SP8 Multiphoton microscope with a coherent laser. A wavelength of 880 nm was optimal for imaging both microglia and the blood vessel dye. The power output at the brain was maintained at 25 mW or below. Images were collected at a 1024 × 1024 pixel resolution using a 25 × 0.9 NA objective with a 1.5× optical zoom. Several fields of view of z-stack images were collected every 1–2 μm through a volume of tissue and used for analysis. To observe microglial dynamics, z-stack time-lapse images were acquired every minute at 2 μm steps in depth. For laser injury, the laser's power was adjusted to 250 mW for 1 s at 880 nm wavelength and ×48 magnification.

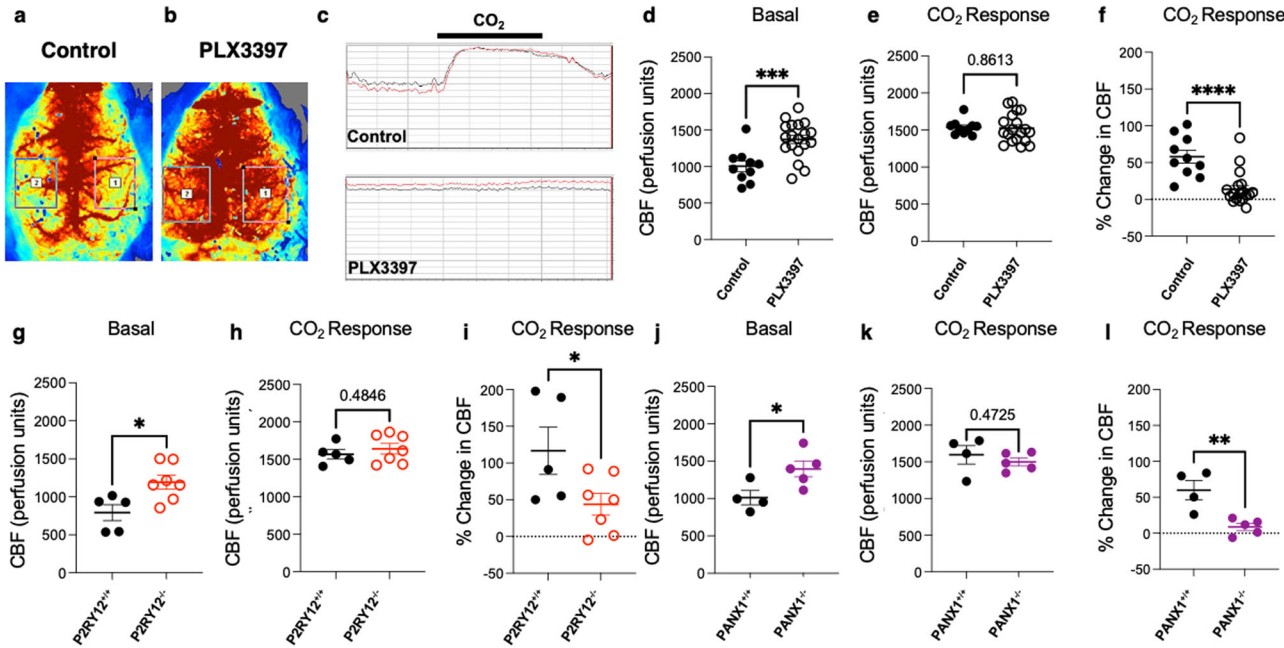

**Fig. 7 Microglia regulate vascular function through PANX1–P2RY12 coupling. a–c** Representative laser speckle images (**a**, **b**) and graphical tracing (**c**) of cerebral blood flow in control and following PLX3397 treatment. **d–f** Quantification of basal cerebral blood flow (**d**), cerebral blood flow in response to $CO_2$ response (**e**), and the change in cerebral blood flow with $CO_2$ treatment (**f**) in control or PLX3397-treated conditions. $n = 10$ control and 19 PLX3397-treated mice. **g–l** Quantification of basal cerebral blood flow (**g**, $n = 5$-7 mice each and **j**, $n = 4$-5 mice each) cerebral blood flow in response to $CO_2$ response (**h**, $n = 5$-7 mice each and **k**, $n = 4$-5 mice each) and the change in cerebral blood flow with $CO_2$ treatment (**i**, $n = 5$-7 mice each and **l**, $n = 4$-5 mice each) in P2RY12$^{+/+}$ and P2RY12$^{-/-}$ mice (**g–i**) and PANX1$^{+/+}$ and PANX1$^{-/-}$ mice (**j–l**) ($n = 4$-7 mice each). Data are presented as mean values ± SEM. *$p < 0.05$; **$p < 0.01$; ***$p < 0.001$; ****$p < 0.0001$. Two-sided unpaired Student's $t$ test.

**Imaging data analysis**. Image analysis was done using ImageJ version 1.53k. CAMs were identified as GFP$^+$ cells having their cell bodies in close physical contact with the vasculature through the two-photon or confocal z-stack. CAMs were identified as ramified myeloid cells with cell bodies florescence overlapping with the capillary fluorescence through the z-stack of the image. To quantify CAMs, total microglia and CAMs were identified in a given stack of images and used to calculate the %CAM in the given field of view. Multiple (3–5) fields of view were used from each animal and the data averaged per animal to ensure a better sampling of the CAM population in a given animal. The dynamics of microglial movement was also analyzed for images collected from different fields of view imaged every week for 4 weeks. For each animal, 3–5 fields of view were chosen based on the clarity and alignment of the imaging over a volume of $120 \times 296 \times 296\ \mu m^3$. The data from the fields of view were averaged and presented per animal. All PCMs and CAMs in the images from the previous week were identified and numbered, and these images were superimposed over images from the following week for the same region being analyzed. Cell bodies were regarded as "stable" if their positions remained the same between the superimposed images. Movement from parenchyma to the vessel was defined as "hop on," whereas the opposite was considered as "hop off." CAM movement along the vessel wall was considered as "crawling."

**Tissue preparation**. For confocal microscopy studies, mice were anesthetized with 5% isoflurane, and transcardially perfused with sodium phosphate buffer (PBS; 50 mM at pH 7.4), followed by 4% paraformaldehyde (PFA). For electron microscopy studies, mice were anesthetized with sodium pentobarbital (80 mg/kg, intraperitoneally) and perfused with 3.5% acrolein, followed by 4% PFA[61]. All perfusion solutions were chilled on ice prior to use. Using a vibratome (Leica VT100S), 50 μm thick sections of the brain were cut in chilled PBS. Slices were then stored in cryoprotectant (40% PBS, 30% ethylene glycol, and 30% glycerol) at −20 °C while further processing took place[29]. Brain sections containing the ventral hippocampus CA1 (Bregma −3.27 and −4.03 in the stereotaxic atlas), the frontal cortex (Bregma 2.93 and –2.57), and sensorimotor cortex (Bregma −2.5 and +2.0) were examined.

**Immunostaining and lectin labeling**. For immunohistochemical staining for light microscopy analysis, brain sections were washed in PBS, blocked with blocking buffer, and incubated overnight at 4 °C with primary antibody solution against IBA1 (1:800, Wako, # 019-19741), CD206 (1:300, BioLegend, # 141701), CD31 (1:150, Millipore, # MAB1398Z), CD13 (0.8 μg/mL, R&D Systems, # AF2335), AQP4 (1:400, Sigma, # AB3594), CSF1R (1:200, Abcam, # ab254357), and P2RY12 (1:300, AnaSpec, # AS-55043A). Sections were then washed thoroughly to remove

excess antibodies and treated with fluorescently tagged secondary antibodies, washed, mounted on slides, and cover-slipped with DAPI mounting medium. The details of the different staining conditions are enclosed in Table 1.

To evaluate the vasculature, a retroorbital injection of DyLight 594-LEL (10 μL Vector Laboratories, # DL-1177) was performed. Briefly, mice were placed in an isoflurane induction chamber, and when fully under anesthesia, mice were placed in lateral recumbency with the eye to be injected facing up. The skin was retracted resulting in the protrusion of the eye. A needle was inserted into the medial canthus at about a 45° angle and 10 μL of the lectin were gently injected. Ten minutes after lectin injection, mice were perfused with ice-cold PBS, followed by 4% PFA. The tissue was subsequently fixed overnight in 4% PFA.

For electron microscopy studies for IBA1 immunolabelling, brain sections were washed in PBS, quenched, and incubated with IBA1 primary antibody (Wako, # 019-19741). Sections were then treated first with goat anti-rabbit secondary antibody conjugated to biotin (Jackson ImmunoResearch, code # 111-066-046) and then with ABC Vectastain system (1:100 in Tris-buffered saline; Vector Laboratories, # PK-6100). To reveal the immunostaining, sections were developed with diaminobenzidine (0.05%) and hydrogen peroxide (0.015%).

**Fluorescence and transmission electron microscopy**. While fluorescently immunolabelled brain sections were imaged at the confocal microscope or the Keyence microscope, and image analysis was done using ImageJ, immunostained electron microscopy sections were postfixed flat in 1% osmium tetroxide and dehydrated in progressively higher concentrations of ethanol. The sections were treated with propylene oxide, impregnated with Durcupan resin (EMS) overnight at room temperature, and then mounted between ACLAR embedding films (EMS). After curing the sheets at 55 °C for 72 h, areas of interest were removed from the embedding films, re-embedded at the tip of resin blocks, and cut to a thickness of 65–80 nm with an ultramicrotome (Leica Ultracut UC7). The ultrathin sections were examined in an FEI Tecnai Spirit G2 transmission electron microscope at 80 kV on bare square mesh grids (EMS). Photographs of the ultrathin sections were taken with an ORCAHR digital camera (10 MP; Hamamatsu) at various magnifications ranging from 440 to 9300. Ultrastructural profiles of brain parenchymal elements were identified according to well-established criteria. Microglia were identified based on previously described ultrastructural characteristics as well as their immunoreactivity for IBA1[29,61].

**RNAscope assay and fluorescence intensity determination**. DyLight 594-Lectin (Vector Laboratories) was retro-orbitally injected into mice. Mice were freshly perfused, brain excised, and fixed in 4% PFA for 24 h at 4 °C. One day later, the brain was immersed in 30% sucrose. The tissue was frozen in the optimal cutting

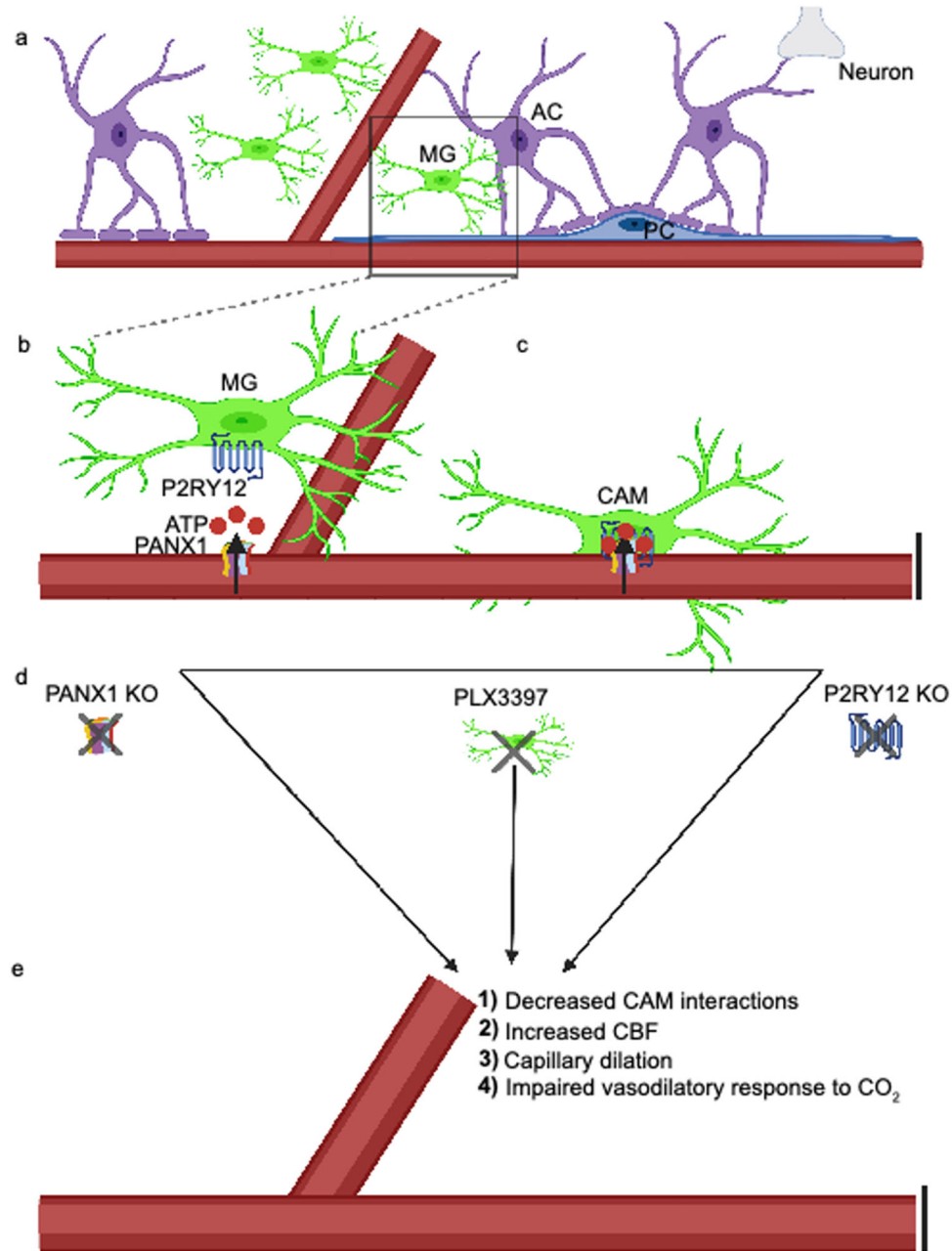

**Fig. 8 Vessel-associated ATP released from pannexin1 (PANX1) attracts microglial cell bodies to capillaries through P2RY12 to regulate homeostatic cerebrovascular physiology.** **a** A cartoon depicting the neurovascular unit consisting of astrocytes (ACs), pericytes (PCs), microglial cells (MGs), neurons, and the associated vasculature. The black box indicates a zoomed-in region depicted in (**b**), showing microglial expression of the P2RY12 and capillary expression of the ATP permeable integral membrane protein PANX1. **c** P2RY12–PAN1 coupling mediate microglial interactions with the vasculature, where those microglia whose cell bodies reside on the vasculature are referred to as capillary-associated microglia (CAMs). Knockout of PANX1, microglial depletion with PLX3397, or knockout of the P2RY12 as depicted in (**d**) all lead to (**e**), reduced CAM interactions, increased capillary diameter and cerebral blood flow, and an impaired vasodilatory response to carbon dioxide ($CO_2$).

temperature embedding media with dry ice. The blocks were sectioned by cutting 15 μm sections. *Sall1* RNA probes were purchased from Advanced Cell Diagnostics (ACD). Here, we used probes against mouse *Sall1* (ACD catalog # 469661-C3), positive control probe (ACD catalog # 310771), and negative probe (ACD catalog # 310043), and then performed the assay by using the RNAscope Fluorescent Multiplex Reagent Kit (ACD catalog # 320850) according to the manufacturer's instructions. Briefly, the fixed frozen tissue slides were postfixed by immersing them in prechilled 4% PFA in 1× PBS for 15 min at 4 °C. We then dehydrated the tissues in 50% EtOH, 70% EtOH, and 100% EtOH each for 5 min at room temperature. Freshly prepared 1× Target Retrieval reagent (ACD catalog # 322000) in a beaker was made and maintained at uniform boiling at 99–100 °C. The slides were then kept in the solution for 3 min. Four drops of Protease III (ACD catalog # 322340) were added to each section, incubated for 30 min at 40 °C, before a

hybridization assay was run. *Sall1* fluorescence intensity was quantified for ten images captured with a ×40 objective lens for each (P2RY12$^{+/+}$; CX3CR1$^{GFP/+}$) mouse. CAMs and PCMs in the cortex were imaged in coronal sections by confocal microscopy with a thickness of 10 μm using an SP8 Lecia system. Analyses of fluorescence intensity were performed using ImageJ version 1.53k.

**Blood vessel diameter analysis**. Two-photon images were collected using the Leica SP8 Multi-photon microscope with the Leica Application Suite X version 3.5.7.23225 software. Confocal images were collected with a Leica TCS SP8 confocal microscope using the Leica Application Suite X version 3.5.5.19976 software. For blood vessel diameter analysis, two-photon images were collected of microglia and capillaries every other day (days 0, 2, and 4) from either control or PLX3397-

treated mice. From collected images, capillaries were selected at random, and their lengths were measured in both conditions on the first (day 0) and fifth day (day 4) of control or PLX3397 treatment. The percent change in the capillary size was determined as a ratio of the capillary size by the fifth day compared to the first day of control or PLX3397 treatment. At least five capillaries from three to five fields of view (11 fields of view from 3 control and 13 fields of view from 3 PLX3397-treated mice) were quantified.

**Morphological comparisons between CAMs and PCMs.** For primary process analysis, the number of primary processes were randomly selected from a stack of images for either CAMs or PCMs and manually counted. At least 50 cells were counted from each animal from several fields of view and averaged. For cell body size analysis, the area around the cell body was drawn in ImageJ for CAM or PCM cells and the area was determined and averaged for at least 50 cells per animal. Individual CAMs and PCMs were identified and using ImageJ, the image was binarized and thresholded for each individual cell. The whole-cell area was quantified and averaged from at least 20 cells per animal. For basal motility, analysis was conducted as previously described[62]. Briefly, as with the whole-cell area, images were binarized through the time-lapse period of 15 min and changes between adjacent time frames were quantified and averaged for the 15 min period. For chemotactic responses, the number of processes that individual CAM and PCM cells showed in response to a laser-induced injury was quantified from four to five responding cells. For responding cells analysis, all cells that responded to the laser-induced injury within a 50 μm radius from the injury were quantified for CAMs and PCMs.

**Blood vessel volume analysis.** A thresholding segmentation was used to segment blood vessels from the surrounding tissue. Using the built-in surface rendering function in IMARIS, the vascular architecture of the tissue was 3D reconstructed, and the percent volume of the vasculature within the tissue volume was calculated by the software. To calculate the length of the vasculature in the given volume, the "skele-tonization with filament object tracing" algorithm in IMARIS was used to mark the centerlines of the vessel network. The algorithm then calculated the vessel length based on the path length of the filamentous centerline depicting the vasculature.

**KA seizures.** Chemoconvulsive seizures were induced in mice using intraper-itoneal injections of KA at 24–27 mg/kg body weight. Control mice were injected with an equal volume of saline that was used as the vehicle to dissolve the KA. Seizures were scored using a modified Racine scale as follows: (1) freezing behavior; (2) rigid posture with raised tail; (3) continuous head bobbing and forepaws shaking; (4) rearing, falling, and jumping; (5) continuous occurrence of level 4; and (6) loss of posture and generalized convulsion activity[63,64]. Only mice that pro-gressed to stage 4/5 were used for seizure experiments.

**Laser speckle contrast imaging.** For laser speckle imaging, the skin on top of the skull was opened to expose the underlying skull and CBF imaging using the MoorFLPI-2 imaging system (Moor Instruments) was conducted to observe cerebral blood flow changes. During the experiment, CBF was measured in both brain hemispheres (minimum area size for CBF quantification: $3 \times 3$ mm$^2$) for at least 5 mins. CBF is visualized using 16-color bands of perfusion units, and the absolute CBF perfusion units averaged across both hemispheres for mice were measured and used to compare baseline CBF. Mice were also monitored for their CBF in induced hyperemia ($CO_2$ challenge) conditions following baseline CBF recordings for 5 min, after which normal conditions were restored[65]. For calculating the change in CBF, the difference between the mean CBF during hyperemia and the average CBF at baseline was normalized to the CBF at baseline, expressed as % change.

**Statistical analysis.** Student's $t$ test was used to compare two groups. Other com-parisons were evaluated using one-way analysis of variance (more than two groups), followed by post hoc Tukey's test for multiple comparisons within tested groups.

**Reporting summary.** Further information on research design is available in the Nature Research Reporting Summary linked to this article.

## Data availability
The datasets generated and analyzed in this study are available from the corresponding author upon reasonable request. Source data are provided with this paper.

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

## Acknowledgements

We thank members of the Eyo lab and the Center for Brain Immunology and Glia (BIG) for valuable discussions in the development of this project. Graphical illustrations in Figs. 3k and 8 were made using BioRender (https://biorender.com/). This work was supported by The National Institutes of Health (R21NS119727 and R01NS122782; awarded to U.B.E.; 5R01HL137112 and 5P01HL120840 awarded to B.E.I.) and The Owens Family Foundation (awarded to U.B.E.). B.C. and W.A.M. III were supported by the National Institutes of Health Basic Cardiovascular Research Training Grant (5T32HL007284).

## Author contributions

K.B., K.A.O. and K.S. contributed equally to the study by performing experiments and analysis. D.H.L., W.A.M. III, Y.-Y.S. and H.-R.C. also contributed some experiments and analysis to the project. J.O.U., S.A., Z.C., A.B.C.-S., B.C., L.J., J.B. and B.F. contributed analysis to the project. M.-E.T., K.B. and K.S. provided EM analysis. B.E.I., C.-Y.K. and U.B.E. contributed equipment and resources for the study. W.A.M. III, M.-E.T., B.E.I., C.-Y.K. and U.B.E. contributed reagents and resources for the project. K.B., K.A.O., K.S., D.H.L., H.-R.C., W.A.M. III, M.-E.T., B.E.I., C.-Y.K. and U.B.E. contributed to the writing and editing of the manuscript. U.B.E. oversaw the project.

## Competing interests

The authors declare no competing interests.
