## [Peer Review File · Nature Communications]

Reviewers' Comments:

Reviewer #1:

Remarks to the Author:

In this manuscript entitled "Capillary-associated microglia regulate vascular structure and function", Bisht and colleagues report and characterize a brain-resident macrophage subtype ("capillary-associated microglia") that localizes in close proximity to capillaries of distinct brain regions including the hippocampus, cortex and thalamus, but appears functionally and phenotypically identical to parenchymal microglia. Indeed, by using protein markers combined with GFP (CX3CR1) immunoreactivity in Cx3cr1GFP/WT mice, these cells were found to express high levels of pan-microglia markers P2Y12 and CX3CR1 and are negative for the perivascular macrophage marker CD206, suggestive for a parenchymal microglia origin. Consisting of around 30% of total (whole-brain) microglia population, these cells show subtle morphological differences compared to parenchymal microglia through smaller processes but larger cell body as identified by 2-photon imaging, yet do not seem to be functionally different from microglia with respect to their vascular niche of residence. Indeed, CSF1-R inhibitor treatment (PLX3397), eliminating both microglia and capillary-associated microglia, but not affecting density of the latter, resulted in increased capillary diameter after 4-day treatment. Further, these cells are not affected by kainic acid seizures (affecting neuronal activity), nor does P2ry12 or Trem2 genetic deficiency affect dynamic interactions with the microenvironment and density in their residential niche. Together, despite being morphologically slightly distinct, these capillary-associated microglia appear to be unique to their parenchymal counterparts.

Over the years, many studies have elegantly showed microglia diversity using a broad array of microscopic and transcriptomic/proteomic techniques, suggesting the presence of different microglia subsets in distinct spatial and temporal settings. As such, microglia and brain-resident macrophage nomenclature is being updated very regularly based on ontogeny, morphology, gene expression pattern, location, function, and plasticity, in order to benefit the scientific community with a comprehensive classification system to study these cells in health and disease. On the other hand, inaccurate introduction of new nomenclature without clean and meaningful dividing lines with prior characterized subsets might introduce confusion within the field. In the current manuscript, the authors attempt to classify a subset of microglia/brain-resident macrophages residing in a distinct microenvironmental niche, yet provide only few arguments to why these cells would be classified as capillary-associated microglia distinct from parenchymal microglia that just happen to reside within the vascular niche. Citing the authors on pg.8: "CAMs are distinguished from "parenchymal microglia" (PCMs) by their position", but in my opinion, this is not sufficient to give these cells a different name. Of note, transcriptionally distinct vessel-associated macrophages occupy many tissues and organs including the lung, heart, skin, adipose tissue and gut (Chakarov, Science 2020), as such there is no reason to think that the brain will obey different rules. Finally, are these the same cells described by Haruwaka and colleagues ("Vessel-associated microglia" Nat Comm 2019)? Please find major and minor concerns below:

Major concerns are:

-Foremost, this study lacks proper and comprehensive molecular characterization of the cells of interest. If these capillary-associated microglia occupying a unique vascular niche with is distinct from parenchymal microglia in their (mostly) neuronal niche, it is to be expected that enriched transcripts include any related to blood vessel morphology or angiogenesis, for example. Additionally, such molecular analysis could potentially reveal marker genes for further exploration of these cells. Because it is currently impossible to distinguish capillary-associated microglia from parenchymal microglia, and single-cell RNA sequencing studies do not suggest the presence of a transcriptionally distinct microglia population related to vasculature (Masuda et al., Nature 2019), the authors could use an in situ approach to assess this (for example RNAscope HiPlex that allows for multiple gene testing). A careful and thoughtful assessment of microglia markers and comprehensive lack of perivascular macrophage expression are needed as mRNA profiling in situ is possible to distinguish between the two.

-Related to this, if capillary-associated microglia are functionally and phenotypically identical similar to microglia, it might be of interest to investigate their ontological origin as well as their dependency on certain transcriptional factors, for example Myb, Irf8, Runx1.

-The manuscript lacks any mechanism that might explain the interaction between capillary-associated microglia and the vasculature. In this perspective, I appreciate the assessment of P2Y12 and TREM2 as candidate molecules, but I see little rationale as to why these genes might be involved in communication with blood vessels, and find the reasoning behind the experiments performed in the last paragraph (Pg. 10) rather hasty and unclear. In fact, the observed increase in basal cerebral blood flow is an interesting finding, but no evidence is provided that this is due to any alteration in microglia or capillary-associated microglia functioning. Moreover, is Trem2 expressed by these cells? As such, the provided data on vascular function and regulation of capillary diameter by all microglia (Figures 6 and 7) stands alone compared to the description of capillary-associated microglia in Figures 1-3.

-Most importantly, any mechanism or molecular players involved in the crosstalk of these cells with capillaries would greatly strengthen the paper.

Minor points:

-Grammatical and spelling/ writing errors are often found throughout the manuscript. Few examples include:

- Page 7 and 8, the exact same paragraph is repeated twice ("Interestingly, CAM density was increased with a chronic window implantation...").
- Page 8: A small population of microglia (~5-8%) in (?) migrate daily.
- Page 2 and further: Trem2 should be Trem2 unless they mean the protein TREM2
- Page 17: Florescence is Fluorescence?

-To label the vasculature, the authors make use of lectin immunoreactivity. However, I cannot find the procedure back in the materials and methods section of the manuscript. Do the authors make use of lectin intravascular injections? Please include.

-Page 6, Typically, these cells lacked ramified processes (unlike P2RY12+ cells) and expressed lower GFP levels (arrowhead in Supplementary Fig. 2a-c). The cells described by the authors are very likely to be perivascular macrophages. Are these CD206/Lyve1-positive?

- Kainic acid is known to induce vascular changes, as such a direct effect of KA on blood vessels and capillary-associated microglia cannot be ruled out. Please provide evidence that after KA treatment, the effect on vasculature integrity is minimal (i.e. permeability assessment)?

- The manuscript often includes bold and inappropriate statements. For example, the authors claim that density of the capillary-associated microglia is intrinsically regulated (Pg 10 and discussion), however there is no evidence to claim this. To me, intrinsically regulation means self-maintenance without any contribution from peripheral replenishment. The authors could assess the 'intrinsic regulation' of the cells by BrdU pulse-chase labeling after depletion and upon repopulation. In addition, the authors could further speculate/ hypothesize whether there are any environmental signals produced by capillaries that influence capillary-associated microglia replenishment. What is the distribution of CSF1 and/or IL-34 on and around the capillaries, for example? Related to this, please remove the claim on Pg.7 "The vasculature occupied $6.1 \pm 0.93\%$ of the total brain volume whereas... indicating that CAMs are 5X more enriched on the vasculature above what would be expected at random".

Reviewer #2:

Remarks to the Author:

This is a beautiful paper! The origin of this manuscript is a very clever idea to look at the microglia associated with the vasculature and to investigate the implications of this association and functional interactions. I would have expected that the cell bodies of microglia were randomly associated with blood vessels as the microglia tile the parenchyma. However, the authors in this manuscript provide compelling evidence that microglia vascular interactions result in neurovascular alterations and even more surprisingly that this is regulated by microglia Trem2 receptors. The older literature that is cited on macrophage/microglia contributions to vascular development are

not widely known and this study provides a modern context and mechanistic basis for this interaction. There are numerous implications of these findings that will guide further studies on microglia-vascular interactions. This is a terrific study and the manuscript is well-written needing only a few minor edits.

The comparisons between astrocyte and microglia are interesting but not exactly clear as it is described. Numerous studies have shown that astrocytes contact all of the blood vessels on the brain principally with their endfeet whereas their cell bodies are more typically at a distance. Therefore the section on page 5 second last paragraph is not explained clearly enough and may mislead readers who are not aware of the astrocyte literature. The authors are describing the probability that the microglia soma contacts the capillary versus the astrocyte soma. This is a reasonable measurement but it is not a reflection of an increased capillary association of microglia compared to astrocytes. It is a comparison of the contact between capillary and the cell bodies of the 2 cell types. Therefore the sentence "Capillary-associated myeloid cell density was significantly greater than capillary-associated astrocyte density on capillaries in the cortex in vivo" would be accurate if it is revised to "The density of microglia cell bodies associated with capillaries was significantly greater than the association of astrocyte cell bodies with capillaries in the cortex in vivo".

The electron microscopic images of microglia at capillaries is an excellent addition. There is one point that the authors should explicitly clarify. Do the microglia processes and/or soma directly contact the basement membrane (or endothelial cells) with no intervening astrocyte processes? It looks like that is the case but it would help if the authors specifically comment on this because it is not clearly obvious. The direct apposition of microglia soma and/or processes on capillaries will strengthen the argument it is due to direct microglia basement membrane/endothelial cell adhesion.

The authors provide intriguing data that neurovascular coupling is altered by microglia depletion. One possibility is that microglia contribute directly to neurovascular coupling. However they should also provide a caveat particularly in the discussion, that the microglia may indirectly alter neurovascular coupling by modifying the actions of an intermediary (e.g. pericyte, astrocyte, endothelial cell, inhibitory interneuron).

Minor edits

At the start of page 8 there is a section that was mistakenly repeated and has to be corrected. 3 lines from the bottom of page 8- This is an incomplete sentence.

Reviewer #3:

Remarks to the Author:

Capillary-associated microglia regulate vascular structure and function

By Bisht et al

In the manuscript by Bisht et al the authors examine the interactions and relationship of capillary-associated microglia (CAMs) with the vasculature. Using in vivo imaging and numerous histological analysis techniques, the authors emphasize the importance of P2RY12 receptors in CAMs-vasculature interactions and investigate the role of CAMs in capillary dilation and blood flow regulation. They also demonstrate, by using transgenic mouse models, that TREM2 plays an important role in cerebrovascular perfusion.

This is an interesting study that illustrates the importance of microglia in the neurovascular unit. The manuscript would benefit by addressing the following comments:

Major:

1. For the comparison of perivascular astrocyte densities to microglia densities, the authors should

be more cautious in their comparison. Astrocytes tend to have larger territories than microglia, so their densities (measured via cell somas), and particularly capillary-associated densities, will naturally be different. Also, it is more uncommon for astrocyte somas to reside near the vasculature, but astrocyte endfeet are well known to cover approximately 90% of the blood vessel capillary wall as well as of arteriolar walls etc. Therefore, the authors should choose a different perivascular cell type or structure to compare against, such as pericytes, which should be ideal as they cover approximately 70-80% of capillary endothelium (see e.g. *Neuron*. 2010 Nov 4;68(3):409-27, *Nature*. 2012 May 16;485(7399):512-6, *Nat Neurosci*. 2019 Jul;22(7):1089-1098, *J Exp Med*. 2021 Apr 5;218(4):e20202207), or astrocyte endfeet.

2. Based on the observation that CAM did not change after kainic acid-induced seizures, which apart from altering neuronal activity, may also promote neuronal death in the hippocampus. Do the authors believe that this happened because of a short observational window (48h)? Or do they believe that CAM are affected in a region-specific manner?

3. Did depletion of microglia or P2RY12 KO result in any changes on mural cell (pericyte, smooth muscle cell) numbers or coverage? Or astrocyte endfoot coverage? Pericytes are known to regulate vessel diameter and blood flow at the capillary level (see e.g. *Nature*. 2014 Apr 3;508(7494):55-60, *Nat Neurosci*. 2017 Mar;20(3):406-416, *Nat Neurosci*. 2019 Jul;22(7):1089-1098, *Front Cell Neurosci*. 2020 Feb 14;14:27) along with astrocytes at the capillary level (*Nat Neurosci*. 2016 Dec;19(12):1619-1627, *J Neurosci*. 2016 Sep 7;36(36):9435-45). The findings should be discussed in the light of these studies.

4. Related to the previous comment, it has been reported that pericytes under certain conditions could give rise to microglia (*J Cereb Blood Flow Metab* 2006; 26: 613-624, *Acta Neuropathol* 2014; 128: 381-396, *J Neuroinflammation* 2016; 13: 57). Could this have an effect on both pericyte coverage, which would affect blood brain barrier stability and vascular tone/neurovascular coupling, and microglial populations?

5. Did depletion of microglia or P2RY12 KO result in any blood-brain barrier disruption or leakage?

6. In Fig4a, microglial cells appear activated after repopulation (thick and short processes). Can you please discuss and explain?

7. Please provide more details on how the data is quantified and statistical quantification. For example, for cell quantifications, was data quantified by ROI field, by mouse, or etc? What are the n-numbers for each experiment and figure panel?

8. What might be the mechanisms involved in microglial regulation of capillary diameter? At minimum, discussion is needed.

Minor:

9. Last paragraph on page 7 (beginning of page 8) has been copied twice. Please delete duplicate.

10. In Fig2c, the lectin and cx3cr1-gfp images are rather blurry. Could you please substitute with better reps?

11. Some figure panel quantifications are missing error bars.

12. On page 8, the text indicates that the "hop off" microglia data is not shown. However, it is quantified in figure 3. The authors should correct the reference to this information.

13. How were mice monitored during imaging sessions? How closely matched were physiological conditions between imaging sessions on the same mice? This can have an influence on vessel diameter that is independent of the microglial manipulations.

REVIEWER COMMENTS

Reviewer #1 (Remarks to the Author):

Over the years, many studies have elegantly showed microglia diversity using a broad array of microscopic and transcriptomic/proteomic techniques, suggesting the presence of different microglia subsets in distinct spatial and temporal settings. As such, microglia and brain-resident macrophage nomenclature is being updated very regularly based on ontogeny, morphology, gene expression pattern, location, function, and plasticity, in order to benefit the scientific community with a comprehensive classification system to study these cells in health and disease. On the other hand, inaccurate introduction of new nomenclature without clean and meaningful dividing lines with prior characterized subsets might introduce confusion within the field. In the current manuscript, the authors attempt to classify a subset of microglia/brain-resident macrophages residing in a distinct microenvironmental niche, yet provide only few arguments to why these cells would be classified as capillary-associated microglia distinct from parenchymal microglia that just happen to reside within the vascular niche. Citing the authors on pg.8: "CAMs are distinguished from "parenchymal microglia" (PCMs) by their position", but in my opinion, this is not sufficient to give these cells a different name. Of note, transcriptionally distinct vessel-associated macrophages occupy many tissues and organs including the lung, heart, skin, adipose tissue and gut (Chakarov, Science 2020), as such there is no reason to think that the brain will obey different rules. Finally, are these the same cells described by Haruwaka and colleagues ("Vessel-associated microglia" Nat Comm 2019)?

Response: We agree with this reviewer that careful nomenclature is important in the field, and we assure the reviewer that we have accepted to provide this nomenclature with care because we would rather be precise than confusing. While we are aware of the Haruwaka study which refers to the "vessel-associated microglia", that work was not specific for the size of the vasculature studied. However, we would expect that our "capillary-associated microglia" should be included in the broader "vessel-associated microglia" group. Nevertheless, our studies were focused on the capillary so we used our term, not as a way to distinguish it from others, but to indicate the precise site in which we focused as we stated in the third sentence of our Results section "We focused on capillaries (ranging from ~5-10µm in diameter) because the capillary bed represents the most elaborate component of the vasculature, the site of oxygen/nutrient delivery and waste uptake and often undergo the most elaborate remodeling". We wanted to avoid any imprecise conclusion that can be drawn from our work which may not hold true e.g. for larger blood vessels such as arteries or veins which we did not examine in detail. Therefore, our use of the term is for precision and clarity. We are glad for the shared agreement between us and this Reviewer for precision and clarity. We hope that the reviewer will consider that ours is a legitimate nomenclature of distinction, much like the disease-associated microglia (DAMs), proliferation-associated microglia (PAMs), tumor-associated macrophages/microglia (TAMs), border associated macrophages (BAMs), vessel-associated microglia (VAMs) or axon initial segment (AXIS) microglia etc. that have recently been introduced into the literature. Like CAMs, the latter three are nomenclature given in light of positional distinctions (without evidence of transcriptional distinctions) from parenchymal microglia which we think warrants positional descriptions/distinctions as we have done with CAMs. We have now included a discussion of this from the bottom of page 13 - 17 of the Discussion.

Please find major and minor concerns below:

Major concerns are:

-Foremost, this study lacks proper and comprehensive molecular characterization of the cells of interest. If these capillary-associated microglia occupying a unique vascular niche with is distinct from parenchymal microglia in their (mostly) neuronal niche, it is to be expected that enriched transcripts include any related to blood vessel morphology or angiogenesis, for example. Additionally, such molecular analysis could potentially reveal marker genes for further exploration of these cells. Because it is currently impossible to distinguish capillary-associated microglia from parenchymal microglia, and single-cell RNA sequencing studies do not suggest the presence of a transcriptionally distinct microglia population related to vasculature (Masuda et al., Nature 2019), the authors could use an *in situ* approach to assess this (for example RNAscope HiPlex that allows for multiple

gene testing). A careful and thoughtful assessment of microglia markers and comprehensive lack of perivascular macrophage expression are needed as mRNA profiling in situ is possible to distinguish between the two.

Response: The reviewer is right that single-cell RNA sequencing studies do not suggest the presence of a transcriptionally distinct microglial population related to the vasculature. We have not claimed that CAMs are “transcriptionally distinct” from parenchymal microglia (PCMs). Indeed, we reported that they express CX3CR1 and P2RY12 akin to all microglia. We are only highlighting a positional (not transcriptional) distinction between parenchymal and capillary-associated microglia. And it is because we have found no transcriptional difference between CAMs and PCMs that we didn’t further investigate transcriptional distinctions. Nevertheless, at the reviewer’s request, we have done some RNAscope experiments for *Sall1*, a microglial-specific homeostatic transcription factor, to certify that CAMs and PCMs show similar *Sall1* transcript levels as shown in **Fig. 3a-b** and the corresponding text in the Results section in the 1st paragraph of page 8.

-Related to this, if capillary-associated microglia are functionally and phenotypically identical similar to microglia, it might be of interest to investigate their ontological origin as well as their dependency on certain transcriptional factors, for example Myb, Irf8, Runx1.

Response: This is very good question / suggestion. While we think this could be addressed, performing these ontogenic studies would require an extensive period of time. Since global KO of these transcripts are often non-viable, conditional KOs will need to be used which would require two rounds of breeding for each transcription factor of interest. Moreover, given our lack of evidence of transcriptional distinctions (e.g. with *Sall1* expression) between CAMs and PCMs, we did not think this was a direction to pursue in the limited time we have. We hope that this reviewer, whose suggestions have significantly improved our manuscript, can understand this given especially that we have attempted to address all the other questions and concerns. We think it is a great suggestion and will consider testing this in future studies.

-The manuscript lacks any mechanism that might explain the interaction between capillary-associated microglia and the vasculature. In this perspective, I appreciate the assessment of P2Y12 and TREM2 as candidate molecules, but I see little rationale as to why these genes might be involved in communication with blood vessels, and find the reasoning behind the experiments performed in the last paragraph (Pg. 10) rather hasty and unclear. In fact, the observed increase in basal cerebral blood flow is an interesting finding, but no evidence is provided that this is due to any alteration in microglia or capillary-associated microglia functioning. Moreover, is Trem2 expressed by these cells? As such, the provided data on vascular function and regulation of capillary diameter by all microglia (Figures 6 and 7) stands alone compared to the description of capillary-associated microglia in Figures 1-3.

-Most importantly, any mechanism or molecular players involved in the crosstalk of these cells with capillaries would greatly strengthen the paper.

Response: We agree with this reviewer that our initial submission was deficient in (i) a mechanism for CAM interactions and (ii) a connection between CAM interactions and the cerebral blood flow (CBF) findings. We consider this the strongest critique of our manuscript and took it very seriously. To address this, we have now tested a Panx1-P2RY12 signaling mechanism providing data that both Panx1 and P2RY12 KOs have reduced CAM interactions and altered/impaired CBF features. We have now included these data as **Fig. 4i-k** and **Fig. 7g-l** as well as the corresponding text in the 1st paragraph of page 9 and the bottom of page 11 to top of page 12. We believe that this has highlighted and significantly improved our manuscript by providing this novel mechanism for both CAM interactions and CBF features. To this end, we have also included a schematic model to summarize our findings as **Fig. 8** and a corresponding mention on page 13.

Minor points:

-Grammatical and spelling/ writing errors are often found throughout the manuscript. Few examples include:
• Page 7 and 8, the exact same paragraph is repeated twice (“Interestingly, CAM density was increased with a chronic window implantation...”).

Response: We have deleted the repetition and tried to proof the manuscript for grammatical and spelling errors.

• Page 8: A small population of microglia (~5-8%) in (?) migrate daily.

Response: We have rectified this by adding “the cerebral cortex” after “in” at the bottom of page 8.

• Page 2 and further: Trem2 should be Trem2 unless they mean the protein TREM2

Response: We have removed all mention of Trem2 from the manuscript.

• Page 17: Florescence is Fluorescence?

Response: Fluorescence. We have corrected this spelling in the Methods subsection on page 24.

-To label the vasculature, the authors make use of lectin immunoreactivity. However, I cannot find the procedure back in the materials and methods section of the manuscript. Do the authors make use of lectin intravascular injections? Please include.

Response: We apologize for this oversight and have now included descriptions of this in Methods subsection in 1st paragraph of page 23.

-Page 6, Typically, these cells lacked ramified processes (unlike P2RY12+ cells) and expressed lower GFP levels (arrowhead in Supplementary Fig. 2a-c). The cells described by the authors are very likely to be perivascular macrophages. Are these CD206/Lyve1-positive?

Response: We have now included data as **Supplementary Fig 2d-e** to show that these P2RY12-negative, non-ramified cells are CD206-positive

- Kainic acid is known to induce vascular changes, as such a direct effect of KA on blood vessels and capillary-associated microglia cannot be ruled out. Please provide evidence that after KA treatment, the effect on vasculature integrity is minimal (i.e. permeability assessment)?

Response: We have now provided evidence using Evan’s Blue staining that the brain cortical BBB is not compromised with KA at 24h or 48h in **Supplementary Fig. 5**.

- The manuscript often includes bold and inappropriate statements. For example, the authors claim that density of the capillary-associated microglia is intrinsically regulated (Pg 10 and discussion), however there is no evidence to claim this. To me, intrinsically regulation means self-maintenance without any contribution from peripheral replenishment. The authors could assess the ‘intrinsic regulation’ of the cells by BrdU pulse-chase labeling after depletion and upon repopulation.

Response: We are sorry for the confusion that was derived from the use of our term. By “intrinsically regulated”, we mean that it is regulated within the brain. We did not perform BrdU pulse-chase labelling after PLX treatment because this has already been done by several studies including the initial study (Elmore et al., 2014) and subsequent studies (Huang et al., 2018) to clearly show that microglial replenishment comes from brain not from outside the brain by infiltrating cells. Thus, to this Reviewer’s concerns, microglial maintenance is indeed done by “self-maintenance”. We have, however, also included a discussion of our use of the term in the second half of page 15.

In addition, the authors could further speculate/ hypothesize whether there are any environmental signals produced by capillaries that influence capillary-associated microglia replenishment. What is the distribution of CSF1 and/or IL-34 on and around the capillaries, for example?

Response: This is an interesting suggestion by the reviewer and we have provided a brief discussion of this in the last sentence of the main paragraph on page 15 of our Discussion. In addition, though not specifically requested by this reviewer, we performed experiments for CSF1R expression on CAMs compared to PVMs and show no differences in protein expression between them and have included this in **Supplementary Fig 6a-b** and the accompanying text at the bottom of page 10.

Related to this, please remove the claim on Pg.7 “The vasculature occupied $6.1 \pm 0.93\%$ of the total brain volume whereas... indicating that CAMs are 5X more enriched on the vasculature above what would be expected

at random”.

Response: We are not exactly sure why this is considered a bold statement. We think this is an important claim based on our findings that has not been previously appreciated and/or reported (to our knowledge). We think it is an accurate claim as well since we find that while the vasculature occupies only 6% of the brain volume, about 30% of the microglia are associated with the vasculature. Our point is that if ~6% of the brain's volume is occupied by blood vessels, then we would expect that of the total microglial population, if placed randomly in the brain, we could expect ~6% of them to associate with blood vessels. Our documentation of ~30% of microglia associating with capillaries in the brain, therefore suggests an enrichment. We have tried to explain this in better detail for clarity in the 1st paragraph on page 7.

Reviewer #2 (Remarks to the Author):

This is a beautiful paper! The origin of this manuscript is a very clever idea to look at the microglia associated with the vasculature and to investigate the implications of this association and functional interactions. I would have expected that the cell bodies of microglia were randomly associated with blood vessels as the microglia tile the parenchyma. However, the authors in this manuscript provide compelling evidence that microglia vascular interactions result in neurovascular alterations and even more surprisingly that this is regulated by microglia Trem2 receptors. The older literature that is cited on macrophage/microglia contributions to vascular development are not widely known and this study provides a modern context and mechanistic basis for this interaction. There are numerous implications of these findings that will guide further studies on microglia-vascular interactions. This is a terrific study and the manuscript is well-written needing only a few minor edits.

The comparisons between astrocyte and microglia are interesting but not exactly clear as it is described. Numerous studies have shown that astrocytes contact all of the blood vessels on the brain principally with their endfeet whereas their cell bodies are more typically at a distance. Therefore the section on page 5 second last paragraph is not explained clearly enough and may mislead readers who are not aware of the astrocyte literature. The authors are describing the probability that the microglia soma contacts the capillary versus the astrocyte soma. This is a reasonable measurement but it is not a reflection of an increased capillary association of microglia compared to astrocytes. It is a comparison of the contact between capillary and the cell bodies of the 2 cell types. Therefore the sentence “Capillary-associated myeloid cell density was significantly greater than capillary-associated astrocyte density on capillaries in the cortex in vivo” would be accurate if it is revised to “The density of microglia cell bodies associated with capillaries was significantly greater than the association of astrocyte cell bodies with capillaries in the cortex in vivo”.

Response: The reviewer is right and we appreciate this important point. We have now included mentions of “cell bodies” throughout the first two paragraphs of the Result section throughout page 5 of the manuscript to clarify this point.

The electron microscopic images of microglia at capillaries is an excellent addition. There is one point that the authors should explicitly clarify. Do the microglia processes and/or soma directly contact the basement membrane (or endothelial cells) with no intervening astrocyte processes? It looks like that is the case but it would help if the authors specifically comment on this because it is not clearly obvious. The direct apposition of microglia soma and/or processes on capillaries will strengthen the argument it is due to direct microglia basement membrane/endothelial cell adhesion.

Response: The reviewer is right again and the microglial soma directly contact the basement membrane with no intervening astrocytic processes. We have now made this point in the 1st paragraph of page 6 while citing a recent study that also documented this (Mondo et. al., 2020)

The authors provide intriguing data that neurovascular coupling is altered by microglia depletion. One possibility is that microglia contribute directly to neurovascular coupling. However they should also provide a caveat particularly in the discussion, that the microglia may indirectly alter neurovascular coupling by modifying the actions of an intermediary (e.g. pericyte, astrocyte, endothelial cell, inhibitory interneuron).

Response: This is a great suggestion and we have included this caveat at the bottom of page 16 and the top of page 17 of the Discussion.

Minor edits

At the start of page 8 there is a section that was mistakenly repeated and has to be corrected.

Response: We have deleted the repeat.

3 lines from the bottom of page 8- This is an incomplete sentence.

Response: We have corrected the sentence and included “the cerebral cortex” at the bottom of page 8.

Reviewer #3 (Remarks to the Author):

Capillary-associated microglia regulate vascular structure and function
By Bisht et al

In the manuscript by Bisht et al the authors examine the interactions and relationship of capillary-associated microglia (CAMs) with the vasculature. Using in vivo imaging and numerous histological analysis techniques, the authors emphasize the importance of P2RY12 receptors in CAMs-vasculature interactions and investigate the role of CAMs in capillary dilation and blood flow regulation. They also demonstrate, by using transgenic mouse models, that TREM2 plays an important role in cerebrovascular perfusion.

This is an interesting study that illustrates the importance of microglia in the neurovascular unit. The manuscript would benefit by addressing the following comments:

Major:

1. For the comparison of perivascular astrocyte densities to microglia densities, the authors should be more cautious in their comparison. Astrocytes tend to have larger territories than microglia, so their densities (measured via cell somas), and particularly capillary-associated densities, will naturally be different. Also, it is more uncommon for astrocyte somas to reside near the vasculature, but astrocyte endfeet are well known to cover approximately 90% of the blood vessel capillary wall as well as of arteriolar walls etc. Therefore, the authors should choose a different perivascular cell type or structure to compare against, such as pericytes, which should be ideal as they cover approximately 70-80% of capillary endothelium (see e.g. Neuron. 2010 Nov 4;68(3):409-27, Nature. 2012 May 16;485(7399):512-6, Nat Neurosci. 2019 Jul;22(7):1089-1098, J Exp Med. 2021 Apr 5;218(4):e20202207), or astrocyte endfeet.

Response: We agree with this reviewer's comments and caution. We have now emphasized our focus on cell body quantification by mentioning “cell bodies” throughout the first two paragraphs of the Result section throughout page 5 of the manuscript. We have also mentioned this in the first sentence of page 15. We understand that astrocyte endfeet cover the vasculature extensively (and microglial processes don't do so as extensively). Our focus on cell bodies is to emphasize that this type of interaction could provide a distinct (from astrocyte) avenue for microglia to interact functionally with the vasculature. According to the Reviewer's suggestion, we have included a mention of the coverage of pericyte cell bodies on the vasculature and cited the recommended paper accordingly at the bottom of page 5.

2. Based on the observation that CAM did not change after kainic acid-induced seizures, which apart from altering neuronal activity, may also promote neuronal death in the hippocampus. Do the authors believe that this happened because of a short observational window (48h)? Or do they believe that CAM are affected in a region-specific manner?

Response: We have now performed additional experiments using fixed tissues to quantify CAM in control at 24h and at 48h of KA seizures in the hippocampus and like the cortex, we do not find any changes in CAM density with this treatment. This is now included as **Fig. 5e-f** and the corresponding text in the middle of page 10. Therefore, at least in the two regions we examined (the cortex *in vivo* and the hippocampus in fixed slices), we did not detect any changes in CAM density as a result of seizures suggesting unlikely region-specificity.

3. Did depletion of microglia or P2RY12 KO result in any changes on mural cell (pericyte, smooth muscle cell) numbers or coverage? Or astrocyte endfoot coverage? Pericytes are known to regulate vessel diameter and blood flow at the capillary level (see e.g. Nature. 2014 Apr 3;508(7494):55-60, Nat Neurosci. 2017 Mar;20(3):406-416, Nat Neurosci. 2019 Jul;22(7):1089-1098, Front Cell Neurosci. 2020 Feb 14;14:27) along with astrocytes at the capillary level (Nat Neurosci. 2016 Dec;19(12):1619-1627, J Neurosci. 2016 Sep 7;36(36):9435-45). The findings should be discussed in the light of these studies.

Response: We performed pericyte staining in PLX3397-treated and P2RY12-deficient mice and did not observe a change in pericyte or astrocytic end feet numbers which we have now included as **Supplementary figure 7**. However, both PLX3397-treatment and a P2RY12 deficiency resulted in increased pericyte coverage as shown below in **Reviewer Fig. 1**. We are including the pericyte density data in the current manuscript (**Supplementary Figure 7**) and the accompanying descriptions on page 12 but request that we exclude the pericyte data coverage from the current manuscript because this is forming the basis of a future study on microglial-pericyte interactions ongoing in the lab. We included this data here as **Reviewer Fig. 1** confidentially. We have also included data showing that astrocyte coverage of the vasculature by AQP4 does not change with PLX treatment or a P2RY12 deficiency and is also presented in **Supplementary figure 7** with the accompanying description on page 12.

4. Related to the previous comment, it has been reported that pericytes under certain conditions could give rise to microglia (J Cereb Blood Flow Metab 2006; 26: 613–624, Acta Neuropathol 2014; 128: 381–396, J Neuroinflammation 2016; 13: 57). Could this have an effect on both pericyte coverage, which would affect blood brain barrier stability and vascular tone/neurovascular coupling, and microglial populations?

Response: This is an interesting suggestion by the reviewer. However, the first JCBFM paper does not indicate that pericytes would give rise to microglia but rather to other Nestin⁺ neural cells and microglia don't typically express Nestin. While the second Acta Neuropathol study does claim to show this transition, this has been documented only in an injury (ischemic) context and there is no other evidence (to our knowledge) that this kind of transition does occur which is why we did not consider this possibility. In addition, in our hands, we have performed chronic *in vivo* two photon imaging in double transgenic CX3CR1^{GFP/+}:NG2^{dsRed/+} with rhodamine labelled blood vessels to identify microglia (green) and pericytes (vessel associated magenta cells) during PLX treatment. As can be seen in the figure below, we show that while microglial numbers are reduced with the PLX treatment, the pericyte (white arrows) numbers are maintained without any loss in numbers or "transitions". We

include this here confidentially for this reviewer as Reviewer Figure 2 below but not in the manuscript proper because we are working on a different study that is focused on microglia-pericyte interactions.

5. Did depletion of microglia or P2RY12 KO result in any blood-brain barrier disruption or leakage?

Response: We treated mice of different genotypes and conditions including P2RY12KO mice with Evan's Blue and show that the BBB remained intact as we could not detect cortical extravasation of Evan's Blue. We have incorporated this as **Supplementary Figure 5**

6. In Fig4a, microglial cells appear activated after repopulation (thick and short processes). Can you please discuss and explain?

Response: We think the reviewer means Fig. 6a. The reviewer's observation is right. Others have described this including the first study that used the PLX drug and stated regarding repopulating that "...within 3 days IBA1⁺ cells appear throughout the brain with very different morphologies to resident microglia in control brains. They are much larger, with only short stubby processes". We have now made a mention of this and cited Elmore et al., 2014 PMID: 24742461 at the bottom of page 10 and the top of page 11.

7. Please provide more details on how the data is quantified and statistical quantification. For example, for cell quantifications, was data quantified by ROI field, by mouse, or etc? What are the n-numbers for each experiment and figure panel?

Response: In the original submission, we had stated "To quantify CAM, total microglia and CAM were identified in a given stack of images and used to calculate the %CAM in the given field of view. Multiple fields of view were used from each animal to ensure a better sampling of the CAM population in a given animal". We have now added that "3-5" fields of view were analyzed and that "CAMs were identified as ramified myeloid cells with cell bodies fluorescence overlapping with the capillary fluorescence through the z-stack of the image" at the bottom of page 21 of the Methods section. We have also now included the n for all our experiments in the figure legends.

8. What might be the mechanisms involved in microglial regulation of capillary diameter? At minimum, discussion is needed.

Response: We have now provided a brief discussion of possible regulatory mechanisms on pages 14-16 of the revised manuscript.

Minor:

9. Last paragraph on page 7 (beginning of page 8) has been copied twice. Please delete duplicate.

Response: We have deleted the duplicate.

10. In Fig2c, the lectin and cx3cr1-gfp images are rather blurry. Could you please substitute with better reps?

Response: We have replaced the images in **Fig. 2c** for clearer images

11. Some figure panel quantifications are missing error bars.

Response: We have ensured that error bars are in all our graphs.

12. On page 8, the text indicates that the “hop off” microglia data is not shown. However, it is quantified in figure 3. The authors should correct the reference to this information.

Response: We have removed the mention of “data not shown”. We had mentioned that because we did not show an image example as with the other types of interactions. But we agree that the data is indeed shown in the quantification.

13. How were mice monitored during imaging sessions? How closely matched were physiological conditions between imaging sessions on the same mice? This can have an influence on vessel diameter that is independent of the microglial manipulations.

Response: We matched the conditions as best as possible using littermates exposed to window implantation surgeries at the same time but divided to receive either control or PLX3397-laden chow. Mice in each group were followed on the same day and at the same time. This is a valid concern and the reason why we performed imaging in control-treated mice as well as PLX3397-treated mice. As can be seen from **Fig. 6k**, there is some variability in the changes with specific capillaries: some increase in size and others decrease in size leading to an overall lack of change in capillary when comparing imaging sessions. However, with the PLX3397 treatment, all the capillaries increased in size with some showing a greater increase than others.

Reviewers' Comments:

Reviewer #1:

Remarks to the Author:

The authors have satisfactorily addressed the concerns of this reviewer. I support their microglia/CAM nomenclature that is based on position rather than identification of a new microglia subpopulation, which they clearly clarified in the discussion.

Reviewer #2:

Remarks to the Author:

The authors have satisfactorily revised the manuscript. No further issues.

Reviewer #3:

Remarks to the Author:

"Capillary-associated microglia regulate vascular structure and function through PANX1-P2RY12 coupling"

Bisht et al

The authors have addressed most of the comments. However, the authors still need to adequately address point 7:

"Please provide more details on how the data is quantified and statistical quantification...."

Specifically, the authors still need to better detail how the statistical quantification is performed. While the authors now indicate how many mice are used in experiments, the points in some of the graphs do not match the number of mice indicated in the legends (see e.g. figures 3, 6 and 7). The authors need to indicate if statistical comparisons are performed by animal, or by vessel for example. If data is analyzed by vessel, are the same number of vessels per mouse compared? If not, this could artificially skew the statistics. Similarly, the authors describe in their response to point 7 that for image analysis, 3-5 fields of view were analyzed, but do not indicate if the data was averaged per mouse, or if all the fields were averaged together. Again, since different numbers of fields were analyzed per mouse, this again could skew the statistical analysis. Please update and clarify the statistical methodology used for all data.

REVIEWER COMMENTS

Reviewer #1 (Remarks to the Author):

The authors have satisfactorily addressed the concerns of this reviewer. I support their microglia/ CAM nomenclature that is based on position rather than identification of a new microglia subpopulation, which they clearly clarified in the discussion.

Response: We appreciate the support from this reviewer on our clarification and thank them for their contribution to make our manuscript more precise on many fronts.

Reviewer #2 (Remarks to the Author):

The authors have satisfactorily revised the manuscript. No further issues.

Response: We appreciate the support from this reviewer and their help in making our discussions in the manuscript better.

Reviewer #3 (Remarks to the Author):

“Capillary-associated microglia regulate vascular structure and function through PANX1-P2RY12 coupling”
Bisht et al

The authors have addressed most of the comments. However, the authors still need to adequately address point 7:

“Please provide more details on how the data is quantified and statistical quantification....”

Specifically, the authors still need to better detail how the statistical quantification is performed. While the authors now indicate how many mice are used in experiments, the points in some of the graphs do not match the number of mice indicated in the legends (see e.g. figures 3, 6 and 7). The authors need to indicate if statistical comparisons are performed by animal, or by vessel for example. If data is analyzed by vessel, are the same number of vessels per mouse compared? If not, this could artificially skew the statistics. Similarly, the authors describe in their response to point 7 that for image analysis, 3-5 fields of view were analyzed, but do not indicate if the data was averaged per mouse, or if all the fields were averaged together. Again, since different numbers of fields were analyzed per mouse, this again could skew the statistical analysis. Please update and clarify the statistical methodology used for all data.

Response: We sincerely apologize for this oversight. The Reviewer is right and perceptive to note these defects. We appreciate this correction. We address the concern for each figure separately below:

Fig. 3: We have now provided detailed information of the “n” including number of cells, number of fields of view and number of mice assessed for this Figure on pages 38 and 39 because it varies for each panel.

Fig. 6: For Fig. 6k, we have now presented the data in terms of the animals examined (i.e. 3 each) rather than the fields of view (11-13). We have noted these points in the legend for Fig. 6 at the top of page 40.

Fig. 7: The number of mice in our legends has been corrected (10 control and 19 PLX mice). We mistakenly used numbers from a previous edition that did not reflect the most updated numbers. We have now corrected this incongruence and present the data accordingly for Fig. 7d-f. We have also highlighted the fact that data was averaged from the two hemispheres in the Methods on page 28.

With these edits, we hope that it is now clear to this reviewer and future readers that our analytical comparisons were performed between animals rather than between individual fields of view, cells, or vessels.

|